# ON THE OPTIMIZATION DYNAMICS OF RLVR: GRADIENT GAP AND STEP SIZE THRESHOLDS

## ABSTRACT

Reinforcement Learning with Verifiable Rewards (RLVR), which uses simple binary feedback to post-train large language models, has shown significant empirical success. However, a principled understanding of why it works has been lacking. This paper builds a theoretical foundation for RLVR by analyzing its training process at both the full-response (trajectory) and token levels. Central to our analysis is a quantity called the *Gradient Gap*, which formalizes the direction of improvement from low-reward to high-reward regions of the response space. We prove that convergence critically depends on aligning the update direction with this Gradient Gap. Moreover, we derive a sharp step-size threshold based on the magnitude of the Gradient Gap: below it, learning converges, whereas above it, performance collapses. Our theory further predicts how the critical step size must scale with *response length* and the *success rate*, thereby explaining why practical heuristics such as *length normalization* improve stability and showing that, with a fixed learning rate, the success rate can stagnate strictly below $100\%$. We validate these predictions through controlled bandit simulations and LLM experiments, including training Qwen2.5-Math-7B with GRPO.

## 1 INTRODUCTION

Large language models (LLMs) have recently achieved significant advances through reinforcement learning post-training, which aligns them with complex tasks and preferences (Ziegler et al., 2019; Ouyang et al., 2022; Shao et al., 2024; Team et al., 2025). In particular, Reinforcement Learning with Verifiable Rewards (RLVR) has emerged as a powerful approach for post-training LLMs on tasks where success can be automatically checked (e.g. using a compiler or solver) (Guo et al., 2025). RLVR methods have shown impressive empirical gains by leveraging binary success/failure feedback instead of human judgments, thereby simplifying the RL pipeline. Techniques in this vein (e.g. variants of Proximal Policy Optimization, a.k.a. PPO (Schulman et al., 2017), like GRPO (Shao et al., 2024), DAPO (Yu et al., 2025b), Dr. GRPO (Liu et al., 2025), etc.) eliminate the need for learned reward or value models, relying purely on verifiable outcome signals. This has enabled LLMs to achieve state-of-the-art results on challenging reasoning and code-generation tasks, demonstrating the promise of RLVR-driven fine-tuning.

However, empirical progress in RLVR has far outpaced our theoretical understanding. The optimization process remains largely a black box: we do not fully understand why RL-based post-training works so well, under what conditions it might falter, or how to tune it for stable convergence. Recent PPO-based variants (e.g. GRPO, DAPO, Dr. GRPO) have sprung up to improve training stability, each introducing different heuristics like normalizing gradient updates by the output length or standardizing rewards by the group's variance. Yet it remains unclear which of these design choices truly matter; without a principled basis, their adoption is guided more by intuition than by theory. This gap is especially pronounced given RLVR's sparse binary rewards (each episode yields just a single success/failure bit), which make it difficult to analyze how gradient descent navigates the model's vast parameter space or how the policy's output distribution shifts toward higher-reward answers. In practice, practitioners often resort to trial-and-error for critical hyperparameters and algorithmic choices, where a mis-tuning can destabilize training or even cause catastrophic collapse (e.g., forgetting pre-trained knowledge or converging to trivial outputs). These challenges underscore the need for a rigorous theoretical framework to demystify RLVR's optimization dynamics and reduce reliance on guesswork.

This work establishes such a framework, providing a rigorous theoretical foundation for RLVR in LLM post-training. Our key contributions are:

- **Unified RLVR theory:** We develop a principled framework for RLVR under binary rewards, introducing the *Gradient Gap* to characterize the improvement direction from low- to high-reward responses.
- **Convergence guarantees:** We prove the existence of a sharp step-size threshold that separates stable convergence from divergence, providing clear guidance for safe hyperparameter tuning.
- **Length- and success-aware learning rates:** Our theory shows that the effective learning rate must shrink with output length and adapt to task difficulty, offering a theoretical explanation for the stabilizing effect of heuristics such as length normalization and clarifying why fixed step sizes can cause stagnation.
- **Empirical validation:** We validate our theory through bandit simulations and LLM experiments, including fine-tuning Qwen2.5-Math-7B on GSM8K, DeepScaleR, and DAPO-17k datasets with GRPO, demonstrating close alignment between theory and practice.

## 1.1 RELATED WORKS

Recent efforts in RL-based language model post-training have introduced a family of GRPO-style algorithms that extend or modify Proximal Policy Optimization (PPO) for verifiable feedback settings. GRPO itself eliminates the value critic by estimating advantages from a group of sampled responses, using relative reward normalization instead of a learned baseline (Shao et al., 2024). Building on this idea, DAPO augmented GRPO by decoupling the PPO clipping range and dynamically filtering out cases where all responses in a batch are correct or all are incorrect (Yu et al., 2025b). Dr. GRPO revisits the advantage normalization procedure, arguing that removing length and variance normalizations (i.e. using only a mean baseline) can prevent bias in policy updates (Liu et al., 2025). Additional related papers are discussed in Section A.

In parallel, theoretical work has established convergence guarantees for policy gradient (PG) methods, including REINFORCE and actor-critic algorithms. In finite Markov decision processes with softmax policies, the PG objective often satisfies a PL condition, implying that any stationary point is globally optimal and that vanilla gradient ascent converges at a sublinear rate (Agarwal et al., 2021; Xiao, 2022). Actor-critic methods also achieve provable convergence by using two-timescale updates or pessimistic value estimation (Wu et al., 2020; Zanette et al., 2021). However, extending these guarantees to post-training large language models with verifiable binary rewards remains challenging, as sparse success/failure signals provide very limited gradient information.

## 2 PROBLEM SET-UP

**Language Model.**     We begin with a standard language model parameterized by $\theta \in \mathbb{R}^d$, which defines a conditional distribution $\pi_\theta(\vec{\boldsymbol{o}} \mid q)$ over sequences of tokens $\vec{\boldsymbol{o}} = (o_1, o_2, \ldots, o_{|\vec{\boldsymbol{o}}|})$ given an input prompt/question $q$. Output tokens $\{o_t\}_{t=1}^{|\vec{\boldsymbol{o}}|}$ are drawn from a finite vocabulary $\mathcal{T}$, and the generation process ends when the special end-of-sequence token $o_{|\vec{\boldsymbol{o}}|} = \texttt{EOS}$ is emitted.

The model generates tokens in an *autoregressive* fashion: at every step $t$, the next token $o_t$ is sampled conditioned on the prompt $q$ and all previously generated tokens $\vec{\boldsymbol{o}}_{<t} = (o_1, o_2, \ldots, o_{t-1})$. Formally, $\pi_\theta(\vec{\boldsymbol{o}} \mid q) = \prod_{t=1}^{|\vec{\boldsymbol{o}}|} \pi_\theta(o_t \mid q, \vec{\boldsymbol{o}}_{<t})$. Each conditional distribution is defined by a softmax over token logits $\boldsymbol{h}_\theta(\cdot)$:

$$\pi_\theta(o_t \mid q, \vec{\boldsymbol{o}}_{<t}) \ := \ \frac{\exp\{\boldsymbol{h}_\theta(q, \vec{\boldsymbol{o}}_{\leq t})\}}{\sum_{o' \in \mathcal{T}} \exp\{\boldsymbol{h}_\theta(q, \vec{\boldsymbol{o}}_{<t}, o')\}} \ . \tag{1}$$

**Post-Training: Reinforcement Learning with Verifiable Rewards (RLVR).**     While a supervised language model can generate fluent text, it often struggles to align with task-specific goals such as math reasoning or code generation. Post-training addresses this limitation by adapting the model parameters $\theta$ to align more closely with an external reward signal that captures desirable behavior.

Formally, we assume access to an outcome reward model $r^\star(q, \vec{\boldsymbol{o}})$ that is directly verifiable and assessed at the end of a generated sequence: $r^\star = 1$ if the answer is correct (e.g., a valid proof step

or passing code execution) and $r^\star = 0$ otherwise. The aim of reinforcement learning in this context is to tune the model parameters $\theta$ so as to maximize the expected reward under the current policy:

$$\text{maximize}_{\theta \in \mathbb{R}^d} \quad J(\pi_\theta) \ := \ \mathbb{E}_{q \sim \mathbb{P}(Q), \, \vec{\boldsymbol{o}} \sim \pi_\theta(\cdot|q)} \big[ r^\star(q, \vec{\boldsymbol{o}}) \big] \,. \tag{2}$$

**Policy Gradient.** To optimize $J(\pi_\theta)$, we rely on policy gradient–based methods. At each iteration $t$, the parameters $\theta$ are updated according to

$$\theta_{k+1} \ = \ \theta_k + \eta_k \cdot \boldsymbol{w}_k \,, \tag{3}$$

where $\eta_k \geq 0$ is the learning rate and $\boldsymbol{w}_k \in \mathbb{R}^d$ is a normalized update direction with $\|\boldsymbol{w}_k\|_2 \leq 1$. For clarity, we denote the policy and logit function at step $k$ as $\pi_k := \pi_{\theta_k}$ and $\boldsymbol{h}_k := \boldsymbol{h}_{\theta_k}$.

This generic formulation captures a broad family of post-training algorithms used in RLVR. Representative examples are:

*REINFORCE:* The classical policy gradient method updates parameters in the direction

$$\nabla_\theta J(\pi_k) \ = \ \mathbb{E}_{q \sim \mathbb{P}(Q), \, \vec{\boldsymbol{o}} \sim \pi_k(\cdot|q)} \big[ A(q, \vec{\boldsymbol{o}}) \cdot \nabla_\theta \log \pi_k(\vec{\boldsymbol{o}} \mid q) \big] \,, \tag{4}$$

where the advantage function is given by $A(q, \vec{\boldsymbol{o}}) = r^\star(q, \vec{\boldsymbol{o}}) - \mathbb{E}_{\vec{\boldsymbol{o}}' \sim \pi_k(\cdot|q)}[r^\star(q, \vec{\boldsymbol{o}}')]$. In this case, the update rule $\theta_{t+1} = \theta_k + \alpha \cdot \nabla_\theta J(\pi_k)$ can be rewritten in our generic form by setting $\boldsymbol{w}_k = \nabla_\theta J(\pi_k)/\|\nabla_\theta J(\pi_k)\|_2$ and $\eta_k = \alpha \|\nabla_\theta J(\pi_k)\|_2$. Viewed in this way, Dr. GRPO (Liu et al., 2025) emerges as a variant that replaces the single-sample advantage with a group-wise demeaned version.

*Group Relative Policy Optimization (GRPO):* GRPO has recently become a standard choice for RLVR. The full algorithm incorporates clipping ratios and multi-step updates (see Section B.1). To connect it with the generic policy gradient form, we consider a simplified one-step approximation without clipping. In this case, the gradient direction is

$$\boldsymbol{g}_{\text{GRPO}}(\pi_k) \ = \ \mathbb{E}_{q \sim \mathbb{P}(Q), \, \vec{\boldsymbol{o}} \sim \pi_k(\cdot|q)} \Big[ \frac{A(q, \vec{\boldsymbol{o}})}{\sigma(q)} \cdot \frac{1}{|\vec{\boldsymbol{o}}|} \nabla_\theta \log \pi_k(\vec{\boldsymbol{o}} \mid q) \Big] \,, \tag{5}$$

where the conditional standard deviation $\sigma(q)$ is given by $\sigma^2(q) = \text{Var}_{\vec{\boldsymbol{o}} \sim \pi_k(\cdot|q)}[r^\star(q, \vec{\boldsymbol{o}}) \mid q]$. In practice, GRPO is typically trained with a cosine learning rate schedule, which can be locally treated as a constant step size $\alpha$. Within our generic update rule, this corresponds to setting $\boldsymbol{w}_k = \boldsymbol{g}_{\text{GRPO}}/\|\boldsymbol{g}_{\text{GRPO}}\|_2$ and $\eta_k = \alpha \|\boldsymbol{g}_{\text{GRPO}}\|_2$, so that both the response length $|\vec{\boldsymbol{o}}|$ and reward variability $\sigma(q)$ directly influence the effective step size $\eta_k$.

**Objective.** Our goal in this work is to understand how the choice of update direction $\boldsymbol{w}_k$ and step size $\eta_k$ influences the convergence of RLVR. In particular, we ask: under what conditions can we guarantee convergence, and what design choices may lead to instability or failure modes?

## 3 TRAJECTORY-LEVEL ANALYSIS

In this section, we study the optimization scheme (3) on a single prompt $q$. We take a *trajectory-level* view, where each response $\vec{\boldsymbol{o}}$ is treated as a single unit rather than a sequence of tokens. By abstracting away the internal structure, the analysis becomes simpler yet still revealing. We begin by outlining the key ingredients of this trajectory-level view, then examine both its success modes and failure cases. Although this setup is only a warm-up for the more detailed token-level analysis, it already highlights several nontrivial and illuminating properties of RLVR.

### 3.1 KEY INGREDIENTS: GRADIENT GAP AND GAP ALIGNMENT

Recall that the optimization objective is the *correction rate* of the model $\pi_\theta$ on prompt $q$: $J_q(\pi_\theta) := \mathbb{E}_{\vec{\boldsymbol{o}} \sim \pi_\theta(\cdot|q)} \big[ r^\star(q, \vec{\boldsymbol{o}}) \mid q \big]$. To analyze this, we partition the response space $\mathcal{O}$ into two sets based on the verifiable reward $r^\star(q, \cdot)$:

$$\mathcal{O}_q^+ \ := \ \big\{ \vec{\boldsymbol{o}} \in \mathcal{O} \mid r^\star(q, \vec{\boldsymbol{o}}) = 1 \big\} \quad \text{and} \quad \mathcal{O}_q^- \ := \ \big\{ \vec{\boldsymbol{o}} \in \mathcal{O} \mid r^\star(q, \vec{\boldsymbol{o}}) = 0 \big\} \,, \tag{6}$$

Here $\mathcal{O}_q^+$ represents desirable responses (correct solutions), while $\mathcal{O}_q^-$ contains undesirable ones. Accordingly, $J_q(\pi_\theta) = \mathbb{P}_{\vec{\boldsymbol{o}} \sim \pi_\theta(\cdot|q)} \big[ \vec{\boldsymbol{o}} \in \mathcal{O}_q^+ \big]$ and $1 - J_q(\pi_\theta) = \mathbb{P}_{\vec{\boldsymbol{o}} \sim \pi_\theta(\cdot|q)} \big[ \vec{\boldsymbol{o}} \in \mathcal{O}_q^- \big]$.

**Conditional Policies.** We further define conditional distributions over the positive and negative spaces:

$$\pi_\theta^+(\vec{\boldsymbol{o}} \mid q) = \pi_\theta\big(\vec{\boldsymbol{o}} \mid q, \mathcal{O}_q^+\big) \; := \; \frac{\pi_\theta(\vec{\boldsymbol{o}} \mid q)}{J_q(\pi_\theta)} \cdot \mathbb{1}\{\vec{\boldsymbol{o}} \in \mathcal{O}_q^+\}, \tag{7a}$$

$$\pi_\theta^-(\vec{\boldsymbol{o}} \mid q) = \pi_\theta\big(\vec{\boldsymbol{o}} \mid q, \mathcal{O}_q^-\big) \; := \; \frac{\pi_\theta(\vec{\boldsymbol{o}} \mid q)}{1 - J_q(\pi_\theta)} \cdot \mathbb{1}\{\vec{\boldsymbol{o}} \in \mathcal{O}_q^-\}. \tag{7b}$$

These describe how the model $\pi_\theta$ distributes probability mass within the "good" and "bad" regions, respectively.

**Gradient Gap: A Direction for Improvement.** Using the conditional policies, we measure the expected log-probability gradient / score function in each region:

$$\boldsymbol{g}_q^+(\pi_\theta) := \mathbb{E}_{\vec{\boldsymbol{o}} \sim \pi_\theta^+(\cdot\mid q)}\big[\nabla_\theta \log \pi_\theta(\vec{\boldsymbol{o}} \mid q)\big] \quad \text{and} \quad \boldsymbol{g}_q^-(\pi_\theta) := \mathbb{E}_{\vec{\boldsymbol{o}} \sim \pi_\theta^-(\cdot\mid q)}\big[\nabla_\theta \log \pi_\theta(\vec{\boldsymbol{o}} \mid q)\big]. \tag{8}$$

The difference between them,

$$\boldsymbol{g}_q^+(\pi_\theta) - \boldsymbol{g}_q^-(\pi_\theta), \tag{9}$$

is the *Gradient Gap*. Intuitively, it highlights how the model's parameters should be shifted to favor desirable responses over undesirable ones.

Crucially, the Gradient Gap is directly proportional[1] to the policy gradient given in equation (4):

$$\nabla_\theta J_q(\pi_\theta) \; = \; J_q(\pi_\theta)\{1 - J_q(\pi_\theta)\} \cdot \big(\boldsymbol{g}_q^+ - \boldsymbol{g}_q^-\big). \tag{10}$$

This shows that the Gradient Gap captures the true direction of improvement. Unlike the full policy gradient $\nabla_\theta J_q(\pi_\theta)$, $\boldsymbol{g}_q^+(\pi_\theta) - \boldsymbol{g}_q^-(\pi_\theta)$ is not scaled down by the variability factor $J_q(1 - J_q)$, making it a purer indicator of where to move.

**Gap Alignment: Following the Right Direction.** Consider now the optimization scheme (3). At iteration $k$, define $\boldsymbol{g}_q^+(k)$ and $\boldsymbol{g}_q^-(k)$ under the current policy $\pi_k$. The update vector $\boldsymbol{w}_k$ should ideally align with the improvement direction $\boldsymbol{g}_q^+(k) - \boldsymbol{g}_q^-(k)$.

We measure this alignment by the inner product

$$\Delta\mu_q(k) \; := \; \boldsymbol{w}_k \cdot \big\{\boldsymbol{g}_q^+(k) - \boldsymbol{g}_q^-(k)\big\}. \tag{11}$$

If $\|\boldsymbol{w}_k\|_2 = 1$, this equals $\Delta\mu_q(k) \; = \; \|\boldsymbol{g}_q^+(k) - \boldsymbol{g}_q^-(k)\|_2 \cdot \cos\angle\big\{\boldsymbol{w}_k, \boldsymbol{g}_q^+(k) - \boldsymbol{g}_q^-(k)\big\}$, which depends both on the magnitude of the Gradient Gap and the angle of alignment.

In the convergence analysis, $\Delta\mu_q(k)$ will play a central role. For stable progress we require:

    (i) $\Delta\mu_q(k)$ should be positive and preferably large, ensuring updates move in the right direction.

    (ii) The step size $\eta_k$ should adapt to its scale, preventing over- or under-shooting.

### 3.2 MAIN FINDINGS

We now turn to the central findings of our analysis. Proofs will be deferred to Sections C and D. Before presenting the results, let us impose a mild regularity condition on the policy score function.

**Assumption 1** (Regularity of Trajectory Policy Score). *The policy score function $\nabla_\theta \log \pi_\theta(\vec{\boldsymbol{o}} \mid q)$ behaves regularly with respect to the parameters $\theta$:*

    *(a)* (Boundedness) *There exists a constant $G_\mathrm{o} < \infty$ such that for all $\theta$ and $(q, \vec{\boldsymbol{o}})$,*

$$\big\|\nabla_\theta \log \pi_\theta(\vec{\boldsymbol{o}} \mid q)\big\|_2 \; \leq \; G_\mathrm{o}. \tag{12}$$

    *(b)* (Smoothness) *The policy score function is $L_\mathrm{o}$-Lipschitz continuous with respect to $\theta$:*

$$\big\|\nabla_\theta \log \pi_{\theta'}(\vec{\boldsymbol{o}} \mid q) - \nabla_\theta \log \pi_\theta(\vec{\boldsymbol{o}} \mid q)\big\|_2 \; \leq \; L_\mathrm{o} \cdot \|\theta' - \theta\|_2. \tag{13}$$

Throughout this section, we use the shorthand $J_q(k) \; = \; J_q(\pi_k)$ to denote the performance at iteration $k$.

---

[1] A formal proof of this is found in Section B.2

### 3.2.1 Convergence and Stagnation

Armed with this set-up, we now state our main theorem, which distinguishes between two possible outcomes of learning: *successful convergence* to the optimum, or *stagnation* at a suboptimal performance plateau. The distinction hinges on how well the update directions align with the underlying objective. To formalize this, we introduce the notion of *Cumulative Gap Alignment*,

$$M(K) := \sum_{k=0}^{K-1} [\Delta\mu_q(k)]_+ \, \eta_k, \tag{14}$$

which accumulates the amount of "useful progress" made up to horizon $K$. Intuitively, $M(K)$ grows whenever the update direction is positively aligned with the true objective, and it stagnates when the updates fail to exploit the available signal.

**Theorem 1** (Convergence and Stagnation). *Assume that the step sizes satisfy $\eta_k \leq \frac{1}{2\sqrt{L_o}}$.*

(a) **(Stagnation)** *Consider when $J_q(0) < 1$. If the alignment signal is too weak, in the sense that the cumulative alignment remains bounded $M(K) \leq C_0$ and $\sum_{k=0}^{\infty} \eta_k^2 \leq C_0'/(L_o + 8G_o^2)$, for some constants $0 \leq C_0, C_0' < \infty$, then learning will stall. In this case, the performance remains strictly sub-optimal: $J_q(k) \leq J_q(0)\left(J_q(0) + \exp(C_0 + C_0')\{1 - J_q(0)\}\right)^{-1} < 1$.*

(b) **(Convergence)** *Consider a case where $J_q(0) > 0$. Suppose the step size $\eta_k$ is adapted to the strength of the alignment signal,*

$$\eta_k \leq \frac{[\Delta\mu_q(k)]_+}{2\left(L_o + 8\,G_o^2\right)} \quad \text{where } [\,\cdot\,]_+ = \max(0, \cdot). \tag{15a}$$

*Then the performance is lower-bounded at any horizon $K$ by*

$$J_q(K) \geq \frac{J_q(0)}{J_q(0) + \{1 - J_q(0)\}\exp\left\{-\frac{1}{2}M(K)\right\}}. \tag{15b}$$

*Moreover, if the alignment accumulates indefinitely, $\lim_{K\to\infty} M(K) = +\infty$, then the policy is guaranteed to achieve perfect performance: $\lim_{K\to\infty} J_q(K) = 1$.*

The theorem establishes a clear dichotomy. Convergence is attainable only when update directions exhibit consistent alignment with the underlying objective and the step size is properly scaled to reflect this signal. In the absence of either alignment or adaptive scaling, progress stagnates and the policy remains confined to a suboptimal regime.

**Sketch of Proof.** The key step is the inequality

$$\left| \log\left(\frac{J_q(k+1)}{1 - J_q(k+1)}\right) - \log\left(\frac{J_q(k)}{1 - J_q(k)}\right) - \Delta\mu_q(k)\,\eta_k \right| \leq (L_o + 8\,G_o^2)\,\eta_k^2, \tag{16}$$

which is stated formally in Lemma 1 of Section C.1.1. This inequality shows that $\Delta\mu_q(t)\,\eta_t$ captures the first-order Taylor approximation of the change in log-odds of $J_q$. Summing (16) over iterations and analyzing the resulting terms under different cases reveals that the Cumulative Gap Alignment $M(K)$ governs the value of $J_q$. This establishes the claims in Theorem 1.

### 3.2.2 The Importance of Properly Chosen Step Size $\eta_k$

According to condition (15a) in Theorem 1(b), the step size $\eta_k$ must be carefully scaled to match the gap alignment $\Delta\mu_q(k)$. To illustrate this, we contrast two scenarios: a modest step size yields linear convergence, whereas an overly aggressive one causes failure.

*Linear Convergence Under Proper Scaling.* Suppose that every update direction provides a consistent signal, so that the Gap Alignment $\Delta\mu_q(k)$ is uniformly bounded below. In this case, a properly chosen fixed step size is sufficient to guarantee rapid improvement.

**Corollary 1** (Linear Convergence with a Uniform Gap). *If every update direction $\mathbf{w}_k$ provides a uniform gap, $\Delta\mu_q(k) \geq \Delta\mu_q > 0$ for all $k \geq 0$, then a simple fixed step size $\eta$ satisfying*

$$\eta \leq \min\left\{\frac{\Delta\mu_q}{2\left(L_o + 8\,G_o^2\right)}, \frac{1}{2\sqrt{L_o}}\right\},$$

*drives the error to zero at a linear rate: $1 - J_q(K) \leq \frac{1 - J_q(0)}{J_q(0)}\exp\left\{-\frac{1}{2}\Delta\mu_q\,\eta \cdot K\right\}$.*

*The Perils of Overshooting.* The picture changes sharply when the step size is too large. If condition (15a) in Theorem 1(b) is violated, convergence may break down entirely. The next result shows that even with perfect update directions, learning can collapse under overly aggressive step sizes.

**Theorem 2** (Catastrophic Failure from an Overly Large Step Size). *There exists a problem instance under Assumption 1 with $G_{\mathrm{o}} \geq \sqrt{L_{\mathrm{o}}}$ where the Gap Alignment is uniformly positive, $\Delta\mu_q(k) \geq \Delta\mu_q > 0$ for all $k \geq 0$, yet using an overly large constant step size $\eta_k = \eta$ leads to failure. Specifically, if the step size satisfies*

$$\frac{60\,\Delta\mu_q}{L_{\mathrm{o}} + G_{\mathrm{o}}^2} \;\leq\; \eta \;\leq\; \frac{1}{2\sqrt{L_{\mathrm{o}} + G_{\mathrm{o}}^2}}\,,$$

*where $0 < \Delta\mu_q \leq \frac{1}{120}\sqrt{L_{\mathrm{o}} + G_{\mathrm{o}}^2}$, the policy's performance will strictly **decrease** at every step, ultimately converging to zero: $J_q(k) < J_q(k-1)$ and $\lim_{K\to\infty} J_q(k) = 0$.*

While the numerical constants (e.g., 60, 120) are not sharp, the phenomenon is robust: an oversized step size causes repeated overshooting, pushing the system toward collapse rather than improvement.

**Intuition for the lower bound analysis.** Our convergence analysis (Theorem 1) relies on equation (16), which uses a first-order approximation of the change in log-odds. For the lower bound, however, it is crucial to examine the second-order expansion. To this end, we define conditional variances over the positive (and negative) response space: $\mathrm{Var}^+ := \mathrm{Var}_{\vec{\boldsymbol{o}}\sim\pi_k(\cdot\,|\,q,\mathcal{O}_q^+)}\big[\boldsymbol{w}_k \cdot \nabla_\theta \log\pi_k(\vec{\boldsymbol{o}} \mid q)\big]$. The term $\mathrm{Var}^-$ is defined analogously. The second-order Taylor expansion gives

$$\log\left(\frac{J_q(k+1)}{1 - J_q(k+1)}\right) - \log\left(\frac{J_q(k)}{1 - J_q(k)}\right) \;=\; \Delta\mu_q(k)\,\eta_k + \{\mathrm{Var}^+ - \mathrm{Var}^-\}\cdot\eta_k^2 + \mathcal{O}(\eta_k^3)\,. \quad (17)$$

In our construction, the linear term is always favorable: $\Delta\mu_q(k)\,\eta_k > 0$. The challenge comes from the quadratic term. If the variance over the negative space dominates, $\mathrm{Var}^- > \mathrm{Var}^+$, then for moderately large step sizes the second-order effect can overwhelm the first-order gain, pulling the log-odds downward and decreasing $J_q$.

This phenomenon is not just a theoretical artifact—it is highly plausible in practice. Real-world language models typically face an enormous negative space (many incorrect responses) with high variability, leading to large $\mathrm{Var}^-$. In contrast, the positive space often contains only a few consistent modes, keeping $\mathrm{Var}^+$ relatively small. This imbalance highlights the danger of overshooting: unless the step size $\eta_k$ is carefully calibrated, the variance contribution from the negative space can dominate and derail learning. To ensure both stability and progress, the step size must respect the scale $\eta_k \asymp \Delta\mu_q(k)/(L_{\mathrm{o}} + G_{\mathrm{o}}^2)$.

## 4 TOKEN-LEVEL ANALYSIS

We now move towards a token-level analysis of RLVR, which sharpens the trajectory-level perspective developed earlier. While natural and general for abstract analysis, our analysis in Section 3 overlooks the autoregressive structure of LLMs: responses are generated token by token, with intermediate Chain-of-Thought (CoT) steps shaping the learning dynamics.

At the trajectory level, the regularity conditions in Assumption 1 are imposed on the policy score $\nabla_\theta \log\pi_\theta(\vec{\boldsymbol{o}} \mid q)$ of the entire response $\vec{\boldsymbol{o}}$. However, the score can be decomposed into token-wise contributions: $\nabla_\theta \log\pi_\theta(\vec{\boldsymbol{o}} \mid q) = \sum_{t=1}^{|\vec{\boldsymbol{o}}|} \nabla_\theta \log\pi_\theta(o_t \mid q, \vec{\boldsymbol{o}}_{<t})$, where every token $o_t$ requires a forward pass from the language model and thus carries its own regularity properties. This makes it more natural—and ultimately more powerful—to impose assumptions at the token level. Doing so introduces response length as an explicit factor, which will be central to our analysis. Interestingly, as we will see, it also reveals how the training dynamics adapt to task difficulty under the current policy.

We refine Assumption 1 into the following token-level version.

**Assumption 2** (Regularity of Token Policy Score). *There exist $G_{\mathrm{p}}, L_{\mathrm{p}} \in (0, +\infty)$ such that*

$$\big\|\nabla_\theta \log\pi_\theta(o_t \mid q, \vec{\boldsymbol{o}}_{<t})\big\|_2 \;\leq\; G_{\mathrm{p}} < \infty \quad \text{for all } \theta, \text{ question } q, \text{ response prefix } \vec{\boldsymbol{o}}_{<t} \text{ and token } o_t,$$

$$\big\|\nabla_\theta \log\pi_{\theta'}(o_t \mid q, \vec{\boldsymbol{o}}_{<t}) - \nabla_\theta \log\pi_\theta(o_t \mid q, \vec{\boldsymbol{o}}_{<t})\big\|_2 \;\leq\; L_{\mathrm{p}} \cdot \|\theta' - \theta\|_2\,.$$

In addition, we propose a second key assumption concerning the distribution of response length.

**Assumption 3** (Sub-Exponential Response Length). *There exist constants $T_\infty, T_{\psi_1} \in (0, +\infty)$ such that for every question $q$ and every policy $\pi_\theta$, if $\vec{o} \sim \pi_\theta(\cdot \mid q)$ and $\ell := |\vec{o}|$ denotes the response length, then $1 \leq \ell \leq T_\infty$ almost surely, and $\|\ell\|_{\psi_1} \leq T_{\psi_1}$.[2]*

Assumption 3 characterizes response length: $T_\infty$ bounds the worst case, while $T_{\psi_1}$ reflects the typical scale. It holds $\mathbb{E}_{\pi_\theta}[|\vec{o}| \mid q] \leq T_{\psi_1} \leq T_\infty / \log 2$, so that $T_{\psi_1}$ may be much smaller than $T_\infty$.

With these two assumptions in place, we are ready to present our token-level convergence guarantee. The statement parallels the trajectory-level result, but now incorporates the finer granularity of token-wise dynamics. We retain the key quantities from Section 3.1, namely the Gap Alignment $\Delta\mu_q(k)$ from equation (11), and the Cumulative Gap Alignment $M(K)$ from equation (14).

**Theorem 3** (Convergence at the Token-Level). *Assume $J_q(0) > 0$. If the step size $\eta_k$ is scaled to the strength of the alignment signal,*

$$\eta_k \ \leq \ \min\left\{ \frac{[\Delta\mu_q(k)]_+ / 2}{L_{\mathrm{p}} T_\infty + G_{\mathrm{p}}^2 \min\left\{\frac{T_{\psi_1}}{1 - J_q(k)}, 8\,T_\infty^2\right\}}, \ \frac{1}{2\sqrt{L_{\mathrm{p}} T_\infty + G_{\mathrm{p}}^2 T_{\psi_1}}} \right\}, \tag{18a}$$

*then the performance is guaranteed at any horizon $K$ by*

$$J_q(K) \ \geq \ \frac{J_q(0)}{J_q(0) + \{1 - J_q(0)\} \exp\left\{ -\frac{1}{2} M(K) \right\}} \ . \tag{18b}$$

This result closely mirrors the trajectory-level guarantee but introduces several new elements. The response length parameters $T_\infty$ and $T_{\psi_1}$ now play a direct role, reflecting the cost of token-level granularity. In addition, the factor $(1 - J_q)$ emerges in the step-size condition, linking stability to the current performance level of the policy. In a later discussion, we will examine the implications of condition (18a), with particular attention to how step-size choices manifest in practical algorithms such as GRPO and Dr. GRPO.

Complementing the positive result in Theorem 3, we now show that the step-size scalings with $T_\infty$ and $T_{\psi_1}$ are essentially tight, as established by the token-level analogue of Theorem 2 below.

**Theorem 4** (Catastrophic Failure from an Overly Large Step Size at the Token Level). *There exists a problem instance satisfying Assumption 2 with $G_{\mathrm{p}} \geq \sqrt{L_{\mathrm{p}}}$ where the Alignment Gap is always positive, $\Delta\mu_q(k) \geq \Delta\mu_q > 0$, yet choosing a constant step size $\eta_k = \eta$ that is too large leads to a complete failure of learning. Specifically, if the step size satisfies*

$$\frac{120 \,\Delta\mu_q}{(L_{\mathrm{p}} + G_{\mathrm{p}}^2)\, T_\infty} \ \leq \ \eta \ \leq \ \frac{1}{2\sqrt{(L_{\mathrm{p}} + G_{\mathrm{p}}^2)\, T_\infty}} \ , \tag{19}$$

*where $0 < \Delta\mu_q \leq \frac{1}{240}\sqrt{(L_{\mathrm{p}} + G_{\mathrm{p}}^2)\, T_\infty}$, the policy's performance will strictly **decrease** at every step, ultimately converging to zero: $J_q(k) < J_q(k-1)$ and $\lim_{K \to \infty} J_q(K) = 0$.*

This lower bound confirms that the step-size condition (18a) reflects an intrinsic barrier. Indeed, by treating $(1 - J_q(k))$ as constant and applying the crude bound $T_{\psi_1} \lesssim T_\infty$, the upper limit in (18a) reduces to $\eta_k \lesssim [\Delta\mu_q(k)]_+ / \{(L_{\mathrm{p}} + G_{\mathrm{p}}^2)\, T_\infty\}$, which matches the overshooting threshold in (19) up to constants. This alignment verifies the sharp dependence on response length in step-size selection.

Finally, note that the $(1 - J_q(k))$ factor only influences how fast convergence proceeds toward 1. In the lower bound construction of Theorem 4, $J_q(k)$ is strictly decreasing, so this term behaves like a constant and does not alter the failure guarantee. Hence, it affects the upper bound but not the lower bound.

**Implications in GRPO and Dr. GRPO.** We next examine how the update rules of GRPO and Dr. GRPO (or REINFORCE) fit into our token-level framework. For clarity, we restrict attention to the scaling behavior with respect to $\Delta\mu_q$, $T_{\psi_1}$, $J_q$, and $(1 - J_q)$ under a single prompt $q$.

In this regime, the GRPO gradient from equation (5) simplifies to

$$\boldsymbol{g}_{\mathrm{GRPO}}(\pi_k) \ \asymp \ \mathbb{E}_{\vec{o} \sim \pi_k(\cdot|q)}\big[A(q, \vec{o}) \cdot \nabla_\theta \log \pi_k(\vec{o} \mid q)\big] \big/ \big\{ T_{\psi_1} \sqrt{J_q(1 - J_q)} \big\} \ .$$

---

[2]For a random variable $X$, the $\psi_1$-Orlicz norm is $\|X\|_{\psi_1} := \inf\big\{ a > 0 \,:\, \mathbb{E}\left[\exp(|X|/a)\right] \leq 2 \big\}$. Finiteness of $\|X\|_{\psi_1}$ is equivalent to $X$ being sub-exponential.

The Dr. GRPO (or REINFORCE) gradient takes the form (4). Applying identity (10) gives

$$\boldsymbol{g}_{\mathrm{GRPO}} \asymp T_{\psi_1}^{-1}\sqrt{J_q(1-J_q)}\cdot\left(\boldsymbol{g}_q^+ - \boldsymbol{g}_q^-\right) \quad\text{and}\quad \boldsymbol{g}_{\mathrm{Dr.\,GRPO}} \asymp J_q(1-J_q)\cdot\left(\boldsymbol{g}_q^+ - \boldsymbol{g}_q^-\right).$$

An update step $\theta_{k+1} = \theta_k + \alpha\cdot\boldsymbol{g}(\pi_k)$ for $\boldsymbol{g} = \boldsymbol{g}_{\mathrm{GRPO}}$ or $\boldsymbol{g}_{\mathrm{Dr.\,GRPO}}$ can therefore be interpreted as moving in the direction $\boldsymbol{w}_k = (\boldsymbol{g}_q^+ - \boldsymbol{g}_q^-)/\|\boldsymbol{g}_q^+ - \boldsymbol{g}_q^-\|_2$, with alignment magnitude $\Delta\mu_q = \|\boldsymbol{g}_q^+ - \boldsymbol{g}_q^-\|_2$, and effective learning rates

$$(\text{GRPO})\ \ \eta_k \ \asymp\ \Delta\mu_q\cdot T_{\psi_1}^{-1}\sqrt{J_q(1-J_q)} \quad\text{and}\quad (\text{Dr. GRPO})\ \ \eta_k \ \asymp\ \Delta\mu_q\cdot J_q(1-J_q). \tag{20}$$

On the other hand, condition (18a) in Theorem 3, under the simplification $L_{\mathrm{p}} \ll G_{\mathrm{p}}^2$ and retaining the $T_{\psi_1}/(1-J_q)$ term in the denominator, reduces to

$$(\text{Theorem 3, condition (18a)})\qquad \eta_k \ \lesssim\ \Delta\mu_q\cdot T_{\psi_1}^{-1}(1-J_q). \tag{21}$$

Comparing equations (20) and (21) leads to several insights:

*Gradient gap.* Both GRPO and Dr. GRPO scale proportionally with the gap alignment $\Delta\mu_q$, consistent with the theoretical condition.

*Sequence length.* GRPO exhibits the correct $1/T_{\psi_1}$ scaling, aligning with the theory, offering an explanation for why length normalization empirically stabilizes training. In contrast, Dr. GRPO lacks this normalization.

*Correction rate.* After variance normalization, GRPO overshoots as $J_q \to 1$. We hypothesize that this may explain the observed stagnation of training at a correction rate strictly below 1.

**Sketch of Proof for Theorem 3.** The proof builds on the following refined token-level inequality:

$$\log\left(\frac{J_q(k+1)}{1-J_q(k+1)}\right) - \log\left(\frac{J_q(k)}{1-J_q(k)}\right) \ \geq\ \Delta\mu_q(k)\cdot\eta_k - \left(L_{\mathrm{p}}\,T_\infty + \frac{G_{\mathrm{p}}^2\,T_{\psi_1}}{1-J_q(k)}\right)\cdot\eta_k^2. \tag{22}$$

The formal statement of bound (22) is provided in Lemma 2 in Section C.2. In parallel, we adapt the trajectory-level result (16) to the token setting by taking $G_{\mathrm{o}} = G_{\mathrm{p}}T_\infty$ and $L_{\mathrm{o}} = L_{\mathrm{p}}T_\infty$. We then combine these two bounds, applying whichever is tighter in a given regime. The remaining steps follow the same structure as in Theorem 1(b).

The main technical challenge lies in proving inequality (22). The difficulty is that the Gradient Gap $\boldsymbol{g}_q^+ - \boldsymbol{g}_q^-$ is not a martingale, since it depends on the conditional distributions $\pi_\theta^+$ and $\pi_\theta^-$. To address this, we relate the log moment generating functions of the conditional score functions to those of the unconditional scores, which do form martingales. This step is crucial: it yields the sharp linear dependence on $T_{\psi_1}$ in the $\eta_k^2$ term of (22). Without this refinement, a naive trajectory-level analysis would give only the weaker quadratic dependence $G_{\mathrm{p}}^2\,T_\infty^2$.

## 5 Numerical Experiments

### 5.1 REINFORCE on Contextual Bandits

We consider a contextual variant of the synthetic bandit experiment of Arnal et al. (2025, Section 5.1). contextual bandit with contexts $x \in [0,1]^d$ for $d = 10$. For a set of $N := 100$ arms, we generate linear scores for each context $x$, $\mathbf{s}(x) = \boldsymbol{\beta}^\top x \in \mathbb{R}^N$ for a matrix $\boldsymbol{\beta} \in \mathbb{R}^{d\times N}$, with standard normal entries. $r_y(x) := \arg\max_{y\in[N]}\mathbf{s}(x)$. We use linear logits initialized as $\ell_0(x) := \theta_0^\top x \in \mathbb{R}^N$, for $\theta_0 \sim \mathcal{N}(0, 0.01^2\cdot\mathrm{Id}_{d\times d})$. The policy $\pi_{\theta_k}$ was then initialized as a softmax over $\ell_0$ and the parameters $\theta_k$ were updated according to the REINFORCE exact gradient update at a training context $x_k$ with stepsize $\eta := 0.1$:

$$\theta_{k+1} = \theta_k + \eta\ \mathbb{E}_{y\sim\pi_{\theta_k}(\cdot|x_k)}[(r_y(x_k) - J_{x_k}(\pi_{\theta_k}))\cdot\nabla_\theta\log\pi_{\theta_k}(y\mid x_k)].$$

The training context $x_k$ was selected at random among those (from an initial pool of 100 contexts drawn uniformly from $[0,1]^d$) with intermediate value function $J(x_k) \in [0.2, 0.8]$, following intuitions from curriculum learning for filtering out overly difficult or easy prompts (Zhang et al., 2025).

We construct three plots based on calculating the following for 500 randomly evaluated contexts $x$: the value function $J_x(\pi_{\theta_k})$, per-context cumulative gradient gap $\sum_{i=0}^k[\Delta\mu_x(i)]_+\cdot\eta$, and the relative per-context cumulative gradient gap $\sum_{i=0}^k([\Delta\mu_x(i)]_+ - [\Delta\mu_{x_i}(i)]_+)\cdot\eta$ which measures the discrepancy of the gradient gaps at the training contexts $x_k$.

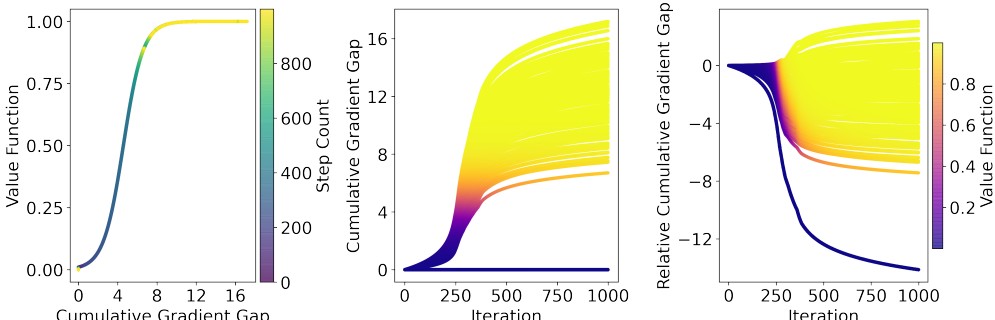

**Figure 1:** Contextual Bandit Experiments.

From the first subplot, we observe a distinctive logistic relationship between the cumulative gradient gap and the value function, reminiscent of our theory (Corollary 1). From the second subplot, we see there are two regimes for each context's cumulative gradient gap curve: either fast exponential convergence (Corollary 1) or lack of improvement (Theorem 2). In the third plot, we interestingly see that those contexts with close to 0 relative cumulative gradient gap (i.e., close to that of training contexts) experience faster convergence.

## 5.2 GRPO ON LANGUAGE MODELS

We validate our theory on three GRPO training runs on the base model Qwen2.5-Math-7B for three math reasoning datasets: (1) the GSM8k dataset (Cobbe et al., 2021), (2) the DeepScaleR dataset (Luo et al., 2025), and (3) the DAPO-17k dataset (Yu et al., 2025a). For background, the GSM8k dataset consists of grade-school math word problems, while the more challenging DeepScaleR and DAPO-17k datasets consist of problems derived from past AIME and AMC competitions.

At each training step, we approximate the batch-average gradient gap magnitude $\mathbb{E}_q[\Delta\mu_q]$ using the relation $\boldsymbol{g}_{\text{GRPO}} \propto \sqrt{J_q(1-J_q)} \cdot (\boldsymbol{g}_q^+ - \boldsymbol{g}_q^-)$, as derived in Section 4. In Figure 2, we plot the cumulative gradient gap vs. the value function, colored by normalized step count. For all three datasets, we see a similar relationship between cumulative gradient gap and accuracy as in our theory and bandit experiment.

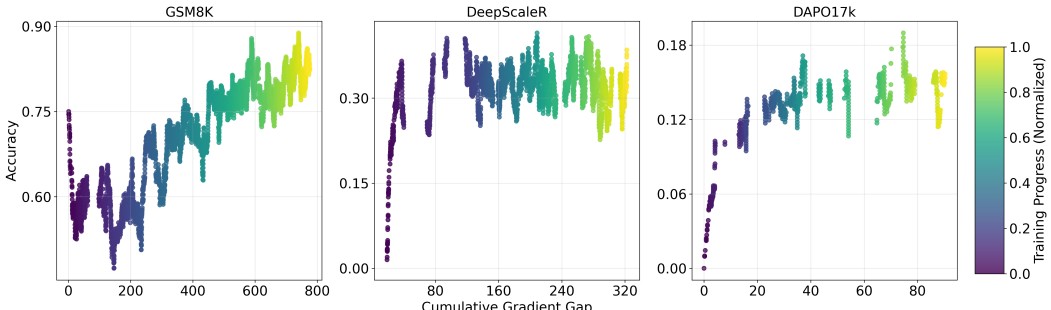

**Figure 2:** Cumulative Gradient Gap vs. Validated Accuracy for our experiments.

## 6 DISCUSSION AND FUTURE DIRECTIONS

Our analysis is restricted to the single-prompt setting, which enabled sharp characterizations of Gradient Gap alignment and step size scaling. In practice, however, training involves a diverse batch of prompts. In this regime, both the alignment signal $\Delta\mu_q(k)$ and the optimal step size $\eta_k$ can vary substantially across prompts. A single update direction $\boldsymbol{w}_k$ may align well with some prompts but poorly with others, and a step size that is safe for one subset may be overly aggressive for another, leading to overshooting and limited overall gains.

These observations suggest several directions for future work: developing prompt-adaptive updates that adjust direction or scale based on batch heterogeneity, analyzing the statistical dynamics of RLVR under diverse prompt distributions, and extending the framework to sequential or curriculum-based training (Bengio et al., 2009; Chen et al., 2025; Zhang et al., 2025). Such extensions are essential for a full theory of RLVR in realistic multi-prompt settings.

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

## THE USE OF LARGE LANGUAGE MODELS

We used large language models (LLMs) solely as an aid for literature review and language refinement. Specifically, LLMs were employed to help identify relevant papers, summarize their contributions, and organize them into coherent thematic categories. No LLM outputs were used in developing theoretical results, experiments, or conclusions.

## A  ADDITIONAL RELATED WORK

A growing body of work has begun to examine the theoretical foundations of preference-based RLHF and verifiable-reward RL. Early studies analyzed preference-driven RL with trajectory-level feedback, establishing convergence guarantees under pairwise or $K$-wise comparisons (Pacchiano et al., 2021; Chen et al., 2022; Zhu et al., 2023). More recent results offered complexity characterizations: Wang et al. (2023) and Du et al. (2024) compared RLHF with standard RL, identifying conditions for sample-efficient preference optimization. In parallel, techniques from function approximation and offline RL have been adapted to the fine-tuning setting (Chen, 2025; Wang et al., 2024; Brantley et al., 2025). Yet the optimization behavior of Reinforcement Learning with Verifiable Rewards (RLVR)—where supervision is provided by deterministic outcomes rather than preferences—remains largely unexplored.

Building on these foundations, researchers have investigated how sparse, outcome-based rewards shape learning dynamics. For example, one study shows that off-policy updates can benefit from emphasizing positive (successful) outcomes more strongly than negative ones (Arnal et al., 2025), while another finds that purely outcome-driven signals can collapse solution diversity in the absence of exploration incentives (Song et al., 2025). These insights illustrate how verifiable binary rewards influence both convergence and the diversity of reasoning strategies, offering early theoretical guidance for RLVR.

Complementing this theoretical line, sequence-level optimization methods have advanced the algorithmic toolkit for RL-based fine-tuning. GSPO, for instance, defines importance ratios over whole-answer likelihoods with sequence-level clipping and updates, improving stability and efficiency compared to token-level methods (Zheng et al., 2025). To control variance from variable output lengths, specialized loss aggregation schemes such as $\Delta L$ normalization have been introduced, yielding an unbiased, minimal-variance estimator of policy loss across different generation lengths (He et al., 2025).

## B  BACKGROUND AND DERIVATIONS

### B.1  REVIEW OF GROUP RELATIVE POLICY OPTIMIZATION (GRPO)

Among post-training algorithms, Group Relative Policy Optimization (GRPO) has emerged as the workhorse for improving LLM reasoning ability toward the verifiable-reward objective $J(\pi)$ in equation (2). The idea is straightforward: instead of evaluating each generated response in isolation, GRPO leverages relative performance within a group.

**Overview of algorithm.**  For each question $q$, we sample a group of $G$ candidate responses $\{\vec{\boldsymbol{o}}^{(i)}\}_{i=1}^{G} \sim \pi_{\theta_{\mathrm{old}}}(\cdot \mid q)$. These raw outcomes are converted into group-normalized advantages:

$$\widehat{A}_t^{(i)} \;=\; \frac{r^{(i)} - \mathrm{mean}(\{r^{(j)}\}_{j\in[G]})}{\mathrm{std}(\{r^{(j)}\}_{j\in[G]})}\,. \tag{23}$$

GRPO then performs a PPO-style update, using a clipped surrogate objective and an optional KL penalty toward a fixed reference policy $\pi_{\mathrm{ref}}$:

$$\mathcal{J}_{\mathrm{GRPO}}(\pi_\theta) \;=\; \mathbb{E}_{q,\{\vec{\boldsymbol{o}}^{(i)}\}} \frac{1}{G} \sum_{i=1}^{G} \frac{1}{|\vec{\boldsymbol{o}}^{(i)}|} \sum_{t=1}^{|\vec{\boldsymbol{o}}^{(i)}|} \Big[ \min\big\{\rho_t^{(i)}\,\widehat{A}_t^{(i)},\, \mathrm{clip}(\rho_t^{(i)}, 1-\epsilon, 1+\epsilon)\,\widehat{A}_t^{(i)}\big\}$$
$$- \beta\, D_{\mathrm{KL}}\big(\pi_\theta(\cdot \mid q, \vec{\boldsymbol{o}}_{<t}^{(i)}) \,\big\|\, \pi_{\mathrm{ref}}(\cdot \mid q, \vec{\boldsymbol{o}}_{<t}^{(i)})\big) \Big], \tag{24}$$

where the density ratio

$$\rho_t^{(i)} \;:=\; \frac{\pi_\theta\big(o_t^{(i)} \mid q, \vec{\boldsymbol{o}}_{<t}^{(i)}\big)}{\pi_{\theta_{\mathrm{old}}}\big(o_t^{(i)} \mid q, \vec{\boldsymbol{o}}_{<t}^{(i)}\big)}$$

corrects for off-policy sampling. In practice, many implementations set $\beta = 0$ and drop the KL term.

**Simplified Form.** To simplify the analysis, we omit clipping and focus on a simplified surrogate:

$$\widetilde{\mathcal{J}}_{\mathrm{GRPO}}(\pi_\theta) \;=\; \mathbb{E}_{q,\{\vec{\boldsymbol{o}}^{(i)}\}} \frac{1}{G} \sum_{i=1}^{G} \frac{1}{|\vec{\boldsymbol{o}}^{(i)}|} \sum_{t=1}^{|\vec{\boldsymbol{o}}^{(i)}|} \rho_t^{(i)} \, \widetilde{A}_t^{(i)}$$

$$= \; \mathbb{E}_{q,\{\vec{\boldsymbol{o}}^{(i)}\}} \frac{1}{G} \sum_{i=1}^{G} \frac{1}{|\vec{\boldsymbol{o}}^{(i)}|} \sum_{t=1}^{|\vec{\boldsymbol{o}}^{(i)}|} \frac{\pi_\theta\big(o_t^{(i)} \mid q, \vec{\boldsymbol{o}}_{<t}^{(i)}\big)}{\pi_{\theta_{\mathrm{old}}}\big(o_t^{(i)} \mid q, \vec{\boldsymbol{o}}_{<t}^{(i)}\big)} \, \widetilde{A}_t^{(i)} \,.$$

with

$$\widetilde{A}_t^{(i)} \;=\; \frac{r^{(i)} - \frac{1}{G-1} \sum_{j \neq i} r^{(j)}}{\sigma(q)} \,. \tag{25}$$

Here $\sigma^2(q)$ denotes the conditional reward variance: $\sigma^2(q) = \mathrm{Var}_{\vec{\boldsymbol{o}} \sim \pi_t(\cdot | q)}[r^\star(q, \vec{\boldsymbol{o}}) \mid q]$.

Evaluating the gradient at $\theta = \theta_{\mathrm{old}}$ yields

$$\nabla_\theta \widetilde{\mathcal{J}}_{\mathrm{GRPO}}(\pi_\theta) \big|_{\theta = \theta_{\mathrm{old}}} \;=\; \mathbb{E}_{q,\{\vec{\boldsymbol{o}}^{(i)}\}} \frac{1}{G} \sum_{i=1}^{G} \frac{1}{|\vec{\boldsymbol{o}}^{(i)}|} \sum_{t=1}^{|\vec{\boldsymbol{o}}^{(i)}|} \frac{\nabla_\theta \pi_{\theta_{\mathrm{old}}}\big(o_t^{(i)} \mid q, \vec{\boldsymbol{o}}_{<t}^{(i)}\big)}{\pi_{\theta_{\mathrm{old}}}\big(o_t^{(i)} \mid q, \vec{\boldsymbol{o}}_{<t}^{(i)}\big)} \, \widetilde{A}_t^{(i)}$$

$$= \; \mathbb{E}_{q,\{\vec{\boldsymbol{o}}^{(i)}\}} \frac{1}{G} \sum_{i=1}^{G} \frac{1}{|\vec{\boldsymbol{o}}^{(i)}|} \sum_{t=1}^{|\vec{\boldsymbol{o}}^{(i)}|} \nabla_\theta \log \pi_{\theta_{\mathrm{old}}}\big(o_t^{(i)} \mid q, \vec{\boldsymbol{o}}_{<t}^{(i)}\big) \, \widetilde{A}_t^{(i)}$$

$$= \; \mathbb{E}_{q,\{\vec{\boldsymbol{o}}^{(i)}\}} \frac{1}{G} \sum_{i=1}^{G} \frac{1}{|\vec{\boldsymbol{o}}^{(i)}|} \nabla_\theta \log \pi_{\theta_{\mathrm{old}}}\big(\vec{\boldsymbol{o}}^{(i)} \mid q\big) \, \widetilde{A}_t^{(i)} \,.$$

Because responses within a group are independent, this simplifies to

$$\nabla_\theta \widetilde{\mathcal{J}}_{\mathrm{GRPO}}(\pi_\theta) \big|_{\theta = \theta_{\mathrm{old}}} = \mathbb{E}_{q, \vec{\boldsymbol{o}} \sim \pi_{\theta_{\mathrm{old}}}(\cdot | q)} \left[ \frac{1}{|\vec{\boldsymbol{o}}|} \nabla_\theta \log \pi_{\theta_{\mathrm{old}}}(\vec{\boldsymbol{o}} \mid q) \cdot \frac{r^\star(q, \vec{\boldsymbol{o}}) - \mathbb{E}_{\pi_{\theta_{\mathrm{old}}}}[r^\star(q, \vec{\boldsymbol{o}}') \mid q]}{\sigma(q)} \right],$$

which is exactly the normalized policy gradient form in equation (5).

## B.2 PROOF OF EQUATION (10)

We begin with the definition of $J_q(\pi_\theta)$, which is the probability of generating a response in the positive space $\mathcal{O}_q^+$:

$$J_q(\pi_\theta) \;=\; \int_{\mathcal{O}_q^+} \pi_\theta(\vec{\boldsymbol{o}} \mid q) \, d\vec{\boldsymbol{o}} \,.$$

First, we take the gradient of $J_q(\pi_\theta)$ and apply the log-derivative trick

$$\nabla_\theta J_q(\pi_\theta) \;=\; \int_{\mathcal{O}_q^+} \nabla_\theta \pi_\theta(\vec{\boldsymbol{o}} \mid q) \, d\vec{\boldsymbol{o}} \;=\; \int_{\mathcal{O}_q^+} \nabla_\theta \log \pi_\theta(\vec{\boldsymbol{o}} \mid q) \, \pi_\theta(\vec{\boldsymbol{o}} \mid q) \, d\vec{\boldsymbol{o}}$$

$$= \; \mathbb{E}_{\vec{\boldsymbol{o}} \sim \pi_\theta(\cdot | q)} \big[ \nabla_\theta \log \pi_\theta(\vec{\boldsymbol{o}} \mid q) \cdot \mathbb{1}\{\vec{\boldsymbol{o}} \in \mathcal{O}_q^+\} \big] \,.$$

This integral can be written as a conditional expectation over the positive response space, which gives us our first identity:

$$\nabla_\theta J_q(\pi_\theta) \;=\; J_q(\pi_\theta) \cdot \mathbb{E}_{\vec{\boldsymbol{o}} \sim \pi_\theta(\cdot | q, \mathcal{O}_q^+)} \big[ \nabla_\theta \log \pi_\theta(\vec{\boldsymbol{o}} \mid q) \big] \;=\; J_q(\pi_\theta) \cdot \boldsymbol{g}_q^+ \,. \tag{26}$$

Alternatively, we can express $J_q(\pi_\theta)$ using the negative response space:

$$J_q(\pi_\theta) \;=\; 1 - \int_{\mathcal{O}_q^-} \pi_\theta(\vec{\boldsymbol{o}} \mid q)\, d\vec{\boldsymbol{o}}\,.$$

This gives us our second identity, this time in terms of the negative response space:

$$\nabla_\theta\, J_q(\pi_\theta) \;=\; -\left\{1 - J_q(\pi_\theta)\right\} \cdot \mathbb{E}_{\vec{\boldsymbol{o}} \sim \pi_\theta(\cdot \mid q, \mathcal{O}_q^-)}\left[\nabla_\theta\, \log \pi_\theta(\vec{\boldsymbol{o}} \mid q)\right] \;=\; -\left\{1 - J_q(\pi_\theta)\right\} \cdot \boldsymbol{g}_q^-\,. \tag{27}$$

From the two identities in (26) and (27), we can isolate the score terms $\boldsymbol{g}_q^+$ and $\boldsymbol{g}_q^-$:

$$\boldsymbol{g}_q^+ \;=\; \frac{\nabla_\theta\, J_q(\pi_\theta)}{J_q(\pi_\theta)} \qquad \text{and} \qquad \boldsymbol{g}_q^- \;=\; -\frac{\nabla_\theta\, J_q(\pi_\theta)}{1 - J_q(\pi_\theta)}\,.$$

Finally, we compute their difference:

$$\boldsymbol{g}_q^+ - \boldsymbol{g}_q^- \;=\; \frac{\nabla_\theta\, J_q(\pi_\theta)}{J_q(\pi_\theta)\{1 - J_q(\pi_\theta)\}}\,.$$

Rearranging this final expression gives the desired result in equation (10), completing the proof.

## C  PROOFS OF CONVERGENCE GUARANTEES

### C.1  TRAJECTORY-LEVEL ANALYSIS

#### C.1.1  PROOF OF THEOREM 1

We begin with the following Lemma 1, which is the core tool for our analysis. The proof is provided in Section E.1.

**Lemma 1.** *Suppose that Assumption 1 holds and the step size*

$$\eta_k \leq \frac{1}{2\sqrt{L_{\mathrm{o}}}}\,.$$

*Then*

$$\left| \log\left( \frac{J_q(k+1)}{1 - J_q(k+1)} \right) - \log\left( \frac{J_q(k)}{1 - J_q(k)} \right) - \Delta\mu_q(k)\, \eta_k \right| \;\leq\; (L_{\mathrm{o}} + 8\, G_{\mathrm{o}}^2)\, \eta_k^2\,. \tag{28}$$

This lemma shows that the single-step improvement is driven by the term $\Delta\mu_q(k) \cdot \eta_k$, with an approximation error proportional to the step size squared.

Iterating inequality (28) over $T$ timesteps gives us a bound on the cumulative effect:

$$\left| \log\left( \frac{J_q(k)}{1 - J_q(k)} \right) - \log\left( \frac{J_q(0)}{1 - J_q(0)} \right) - \sum_{k=0}^{K-1} \Delta\mu_q(k)\, \eta_k \right| \;\leq\; (L_{\mathrm{o}} + 8\, G_{\mathrm{o}}^2) \sum_{k=0}^{K-1} \eta_k^2\,. \tag{29}$$

Using this inequality, we analyze two distinct outcomes.

**(a) Stagnation.**   First, we show that the return $J_q(K)$ can be bounded away from the optimum under the condition from Theorem 1(a). Rearranging the cumulative inequality gives an upper bound:

$$\log\left( \frac{J_q(K)}{1 - J_q(K)} \right) \leq \log\left( \frac{J_q(0)}{1 - J_q(0)} \right) + \sum_{k=0}^{K-1} \Delta\mu_q(k) \cdot \eta_k + (L_{\mathrm{o}} + 8G_{\mathrm{o}}^2) \sum_{k=0}^{K-1} \eta_k^2\,,$$

which further leads to

$$J_q(K) \;\leq\; \frac{J_q(0)}{J_q(0) + \{1 - J_q(0)\} \exp\left\{ - \sum_{k=0}^{K-1} \Delta\mu_q(k)\, \eta_k - (L_{\mathrm{o}} + 8\, G_{\mathrm{o}}^2) \sum_{k=0}^{K-1} \eta_k^2 \right\}}\,.$$

By applying the bounds on the cumulative alignment gap and step sizes from Theorem 1(a), the inequality simplifies to

$$J_q(K) \;\leq\; \frac{J_q(0)}{J_q(0) + \exp(C_0 + C_0')\,\{1 - J_q(0)\}} \;<\; 1\,.$$

This result shows that the performance $J_q(K)$ hits a ceiling and is strictly bounded away from the optimum value of 1, proving the stagnation described in Theorem 1(a).

**(b) Convergence.** Next, we show that an adaptive step size $\eta_k$ guarantees convergence. The key is the rule from condition (15a): If the gap alignment $\Delta\mu_q(k) < 0$, the step size $\eta_k$ is set to zero. This is a "do no harm" policy that prevents any update in the wrong direction. On the other hand, if $\Delta\mu_q(k) \geq 0$, then the step size $\eta_k$ is chosen carefully to ensure that $(L_{\mathrm{o}}+8G_{\mathrm{o}}^2)\,\eta_k^2 \leq \frac{1}{2}[\Delta\mu_q(k)]_+\cdot\eta_k$. This guarantees meaningful progress. The rule (15a) strengthens inequality (28) to a lower bound on progress:

$$\log\left(\frac{J_q(k+1)}{1-J_q(k+1)}\right) - \log\left(\frac{J_q(k)}{1-J_q(k)}\right) \;\geq\; [\Delta\mu_q(k)]_+\,\eta_k - (L_{\mathrm{o}} + 8\,G_{\mathrm{o}}^2)\,\eta_k^2$$

$$\geq\; \frac{1}{2}\,[\Delta\mu_q(k)]_+\,\eta_k\,. \tag{30}$$

By telescoping and rearranging terms in the same manner as before, we obtain:

$$J_q(K) \;\geq\; \frac{J_q(0)}{J_q(0) + \{1 - J_q(0)\}\,\exp\left\{-\frac{1}{2}\sum_{k=0}^{K-1}[\Delta\mu_q(k)]_+\,\eta_k\right\}}\,.$$

This result shows that as the cumulative gap alignment grows, the exponential term shrinks, pushing the performance $J_q(K)$ towards 1. This confirms the convergence guarantee in Theorem 1(b), as described in inequality (15b).

### C.1.2   PROOF OF COROLLARY 1

Suppose $\Delta\mu_q(k) \geq \Delta\mu_q > 0$ for any $k$, and that

$$\eta_k = \eta \;\leq\; \min\left\{\frac{\Delta\mu_q}{2\,(L_{\mathrm{o}} + 8\,G_{\mathrm{o}}^2)},\, \frac{1}{2\sqrt{L_{\mathrm{o}}}}\right\}\,.$$

According to the bound (30), we have by our bound on $\eta_k = \eta$:

$$\log\left(\frac{J_q(k+1)}{1-J_q(k+1)}\right) - \log\left(\frac{J_q(k)}{1-J_q(k)}\right) \;\geq\; \frac{\Delta\mu_q}{2}\cdot\eta\,.$$

Next, rearranging and telescoping in the same manner as the proof of Theorem 1, we obtain

$$1 - J_q(k) \leq \frac{1 - J_q(0)}{1 - J_q(0) + J_q(0)\cdot\exp\left\{K\cdot\frac{\Delta\mu_q}{2}\cdot\eta\right\}} \leq \frac{1 - J_q(0)}{J_q(0)}\cdot\exp\left\{-K\cdot\frac{\Delta\mu_q}{2}\cdot\eta\right\},$$

which completes the proof of Corollary 1.

### C.2   TOKEN-LEVEL ANALYSIS (PROOF OF THEOREM 3)

We begin by providing the formal statement of inequality (22) from the main body. This result, presented below as Lemma 2, serves as the token-level counterpart to the trajectory-level analysis in Lemma 1 and establishes a precise lower bound on the one-step improvement of the log-odds ratio.

**Lemma 2.** *Suppose Assumptions 2 and 3 hold. If the step size $\eta_k$ is sufficiently small so that*

$$\eta_k \;\leq\; \frac{1}{\sqrt{2\,G_{\mathrm{p}}^2\cdot T_{\psi_1}}}\,, \tag{31a}$$

*then the increment of the log-odds satisfies*

$$\log\left(\frac{J_q(k+1)}{1-J_q(k+1)}\right) - \log\left(\frac{J_q(k)}{1-J_q(k)}\right) \;\geq\; \Delta\mu_q(k)\cdot\eta_k - \left(L_{\mathrm{p}}\,T_\infty + \frac{G_{\mathrm{p}}^2\,T_{\psi_1}}{1-J_q(k)}\right)\cdot\eta_k^2\,. \tag{31b}$$

With this lemma, we can proceed to the main proof of Theorem 3. Our strategy is to synthesize the bounds from Lemma 1 (the trajectory-level analysis) and Lemma 2 (the token-level analysis). Our central aim is to demonstrate that for a sufficiently small step size $\eta_k$, the log-odds of the objective function improves at each step according to the following inequality:

$$\log\left(\frac{J_q(k+1)}{1-J_q(k+1)}\right) - \log\left(\frac{J_q(k)}{1-J_q(k)}\right) \;\geq\; \frac{1}{2}\,[\Delta\mu_q(k)]_+\cdot\eta_k\,. \tag{32}$$

To establish this, we consider two distinct cases that cover all possibilities.

**Case I:** $(1 - J_q(k)) T_\infty \geq 1$. In this scenario, we apply the token-level bound from Lemma 2. The lemma guarantees that inequality (32) holds, provided that the step size $\eta_k$ meets conditions

$$\eta_k \;\leq\; \frac{1}{\sqrt{2\,G_{\rm p}^2 \cdot T_{\psi_1}}} \quad \text{and} \quad \left(L_{\rm p}\,T_\infty + \frac{G_{\rm p}^2\,T_{\psi_1}}{1 - J_q(k)}\right) \cdot \eta_k \;\leq\; \frac{1}{2}\,[\Delta\mu_q(k)]_+ \,. \tag{33}$$

**Case II:** $(1 - J_q(k)) T_\infty < 1$. For the alternative case, we fall back on the trajectory-level bound from Lemma 1. By setting the Lipschitz and gradient norm parameters to $L_{\rm o} = L_{\rm p} T_\infty$ and $G_{\rm o} = G_{\rm p} T_\infty$ respectively, the bound (28) ensures that inequality (32) holds if

$$\eta_k \;\leq\; \frac{1}{2\sqrt{L_{\rm p} \cdot T_\infty}} \quad \text{and} \quad \left(L_{\rm p}\,T_\infty + 8\,G_{\rm p}^2\,T_\infty^2\right) \cdot \eta_k \;\leq\; \frac{1}{2}\,[\Delta\mu_q(k)]_+ \,. \tag{34}$$

**Synthesizing the Results.** To ensure inequality (32) holds universally, we must select a condition on the step size $\eta_k$ that satisfies the constraints from both cases. By adopting the more lenient of the two sets of constraints, we arrive at the unified condition specified in (18a).

With this per-step improvement established, the remainder of the proof follows the same structure as the proof of Theorem 1(b). We sum inequality (32) over all iterations from $k = 0$ to $K - 1$, which yields the final convergence result (18b) and completes the proof.

# D  PROOFS OF LOWER BOUNDS

## D.1  TRAJECTORY-LEVEL ANALYSIS LOWER BOUND (PROOF OF THEOREM 2)

For simplicity, we omit the prompt $q$ and consider a fixed instance. The positive set is a singleton, $\mathcal{O}_q^+ = \{\vec{o}_1\}$, while the negative set contains two responses, $\mathcal{O}_q^- = \{\vec{o}_{-1}, \vec{o}_{-2}\}$.

We adopt a linear feature representation for the logits $\boldsymbol{h}_\theta$, defined by

$$\boldsymbol{h}_\theta(q, \vec{o}) \;=\; \langle \boldsymbol{\phi}(q, \vec{o}),\, \theta \rangle\,,$$

with the following feature map:

$$\boldsymbol{\phi}(q, \vec{o}_1) \;=\; \frac{G_{\rm o}}{2}\begin{pmatrix} 0 \\ 1 \end{pmatrix}, \qquad \boldsymbol{\phi}(q, \vec{o}_{-1}) \;=\; \frac{G_{\rm o}}{2}\begin{pmatrix} 1 \\ 0 \end{pmatrix}, \qquad \boldsymbol{\phi}(q, \vec{o}_{-2}) \;=\; \frac{G_{\rm o}}{2}\begin{pmatrix} -1 \\ 0 \end{pmatrix}.$$

By this construction, the Euclidean norm of the score function $\nabla_\theta \log \pi_\theta(\vec{o} \mid q) = \boldsymbol{\phi}(q, \vec{o}) - \mathbb{E}_{\vec{o} \sim \pi_\theta(\cdot \mid q)}\big[\boldsymbol{\phi}(q, \vec{o})\big]$ is uniformly bounded by $G_{\rm o}$ and has Lipschitz constant $L_{\rm o} = 0$.

We now define the optimization scheme:

**Initialization:**

$$\theta_0 \;:=\; \frac{\eta}{3}\begin{pmatrix} -1 \\ 0 \end{pmatrix}. \tag{35}$$

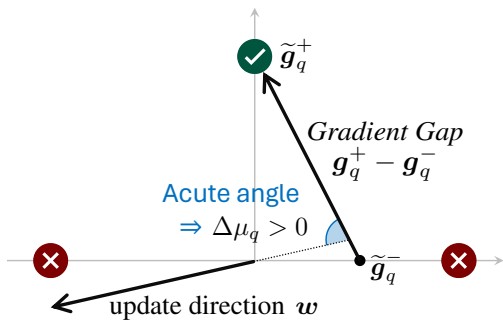

**Figure 3:** Constructed instance for (trajectory-level) lower bound proof.

**Step size:** fixed $\eta_k = \eta$ satisfying

$$\frac{60\,\Delta\mu_q}{L_\text{o} + G_\text{o}^2} \;\le\; \eta \;\le\; \frac{1}{2\sqrt{L_\text{o} + G_\text{o}^2}}\,. \tag{36}$$

**Update direction:**

$$\boldsymbol{w}_k = \frac{1}{3}\begin{pmatrix}(-1)^k \cdot 2 \\ -\delta\end{pmatrix} \qquad \delta := \frac{1}{15}\,\eta\,G_\text{o}\,. \tag{37}$$

The resulting parameter trajectory is

$$\theta_k \;:=\; \frac{\eta}{3}\begin{pmatrix}-(-1)^k \\ -\delta k\end{pmatrix}\,. \tag{38}$$

We can confirm that $\|\boldsymbol{w}_k\|_2 \le 1$ since

$$\frac{4}{9} + \frac{\delta}{9} \le 1 \iff \delta \le 5$$

where the latter inequality is true since $\eta\,G_\text{o} \le \frac{1}{2}$.

Now let us verify two key properties of this trajectory:

**Decreasing value:** The value function $J$ equals the probability assigned to the positive response $\vec{\boldsymbol{o}}_1$, which is given by

$$\begin{aligned}
J_q(k) \;=\; \pi(\vec{\boldsymbol{o}}_1 \mid q) \;&=\; \frac{\exp\big\{-(\delta\eta\,G_\text{o}/6)\cdot k\big\}}{\exp\big\{-(\delta\eta\,G_\text{o}/6)\cdot k\big\} + \exp\{\eta\,G_\text{o}/6\} + \exp\{-\eta\,G_\text{o}/6\}} \\
&=\; \frac{1}{1 + \big\{\exp(\eta\,G_\text{o}/6) + \exp(-\eta\,G_\text{o}/6)\big\}\cdot\exp\big\{(\delta\eta\,G_\text{o}/6)\cdot k\big\}}\,.
\end{aligned}$$

This expression decreases monotonically in $k$.

**Positive gap:** Consider even indices $k$ (the odd case is symmetric). Then

$$\boldsymbol{w}_k = \frac{1}{3}\begin{pmatrix}2 \\ -\delta\end{pmatrix}\,. \tag{39}$$

The conditional probabilities of the two negative responses are

$$\begin{aligned}
\pi_\theta\big(\vec{\boldsymbol{o}}_{-1} \mid q, \mathcal{O}_q^-\big) \;&=\; \frac{\exp(-\eta\,G_\text{o}/6)}{\exp(-\eta\,G_\text{o}/6) + \exp(\eta\,G_\text{o}/6)}\,, \\
\pi_\theta\big(\vec{\boldsymbol{o}}_{-2} \mid q, \mathcal{O}_q^-\big) \;&=\; \frac{\exp(\eta\,G_\text{o}/6)}{\exp(-\eta\,G_\text{o}/6) + \exp(\eta\,G_\text{o}/6)}\,.
\end{aligned}$$

Define

$$\widetilde{\boldsymbol{g}}_q^+ \;:=\; \mathbb{E}_{\vec{\boldsymbol{o}}\sim\pi_\theta(\cdot\mid q,\mathcal{O}_q^+)}\big[\boldsymbol{\phi}(q,\vec{\boldsymbol{o}})\big] \qquad \text{and} \qquad \widetilde{\boldsymbol{g}}_q^- \;:=\; \mathbb{E}_{\vec{\boldsymbol{o}}\sim\pi_\theta(\cdot\mid q,\mathcal{O}_q^-)}\big[\boldsymbol{\phi}(q,\vec{\boldsymbol{o}})\big]\,.$$

Then due to the expression $\nabla_\theta \log\pi_\theta(\vec{\boldsymbol{o}} \mid q) = \boldsymbol{\phi}(q,\vec{\boldsymbol{o}}) - \mathbb{E}_{\vec{\boldsymbol{o}}\sim\pi_\theta(\cdot\mid q)}\big[\boldsymbol{\phi}(q,\vec{\boldsymbol{o}})\big]$, we have

$$\boldsymbol{g}_q^+ - \boldsymbol{g}_q^- \;=\; \{\mathbb{E}^+ - \mathbb{E}^-\}[\nabla_\theta \log\pi_\theta(\vec{\boldsymbol{o}} \mid q)] \;=\; \{\mathbb{E}^+ - \mathbb{E}^-\}[\boldsymbol{\phi}(q,\vec{\boldsymbol{o}})] \;=\; \widetilde{\boldsymbol{g}}_q^+ - \widetilde{\boldsymbol{g}}_q^-\,.$$

Note that

$$\begin{aligned}
\widetilde{\boldsymbol{g}}_q^+ \;&=\; \frac{G_\text{o}}{2}\begin{pmatrix}0 \\ 1\end{pmatrix} \\
\widetilde{\boldsymbol{g}}_q^- \;&=\; \pi_\theta\big(\vec{\boldsymbol{o}}_{-1} \mid q, \mathcal{O}_q^-\big)\cdot\boldsymbol{\phi}(q,\vec{\boldsymbol{o}}_{-1}) + \pi_\theta\big(\vec{\boldsymbol{o}}_{-2} \mid q, \mathcal{O}_q^-\big)\cdot\boldsymbol{\phi}(q,\vec{\boldsymbol{o}}_{-2}) \\
&=\; \frac{1 - \exp(\eta\,G_\text{o}/3)}{1 + \exp(\eta\,G_\text{o}/3)}\cdot\frac{G_\text{o}}{2}\begin{pmatrix}1 \\ 0\end{pmatrix}\,.
\end{aligned}$$

Since $\eta\, G_{\mathrm{o}} \le \frac{1}{2}$, and using the inequality $\frac{e^x - 1}{1 + e^x} \ge \frac{2}{5}x$ for $x \in [0,1]$

$$
\begin{aligned}
\left\langle \boldsymbol{w}_k,\, \boldsymbol{g}_q^+ - \boldsymbol{g}_q^- \right\rangle &= \frac{2}{3} \cdot \frac{\exp(\eta\, G_{\mathrm{o}}/3) - 1}{\exp(\eta\, G_{\mathrm{o}}/3) + 1} \cdot \frac{G_{\mathrm{o}}}{2} - \frac{\delta}{3} \cdot \frac{G_{\mathrm{o}}}{2} \ge \left( \frac{2}{3} \cdot \frac{2}{5}\, (\eta\, G_{\mathrm{o}}/3) - \delta/3 \right) \cdot \frac{G_{\mathrm{o}}}{2} \\
&= \left( \frac{4}{15}\, (\eta\, G_{\mathrm{o}}/3) - \delta/3 \right) \cdot \frac{G_{\mathrm{o}}}{2} = \frac{1}{30}\, \eta\, G_{\mathrm{o}}^2 \ge \Delta\mu_q \,,
\end{aligned}
$$

where the last inequality is true by the lower bound on $\eta$ of (36) and the fact that $L_{\mathrm{o}}^{1/2} \le G_{\mathrm{o}}$. Note that the normalizing constant terms arising in $\nabla_\theta \log \pi_\theta(\vec{\boldsymbol{o}} \mid q)$ within the definitions $\boldsymbol{g}_q^+$ and $\boldsymbol{g}_q^-$ were not computed since they cancel out in the difference $\boldsymbol{g}_q^+ - \boldsymbol{g}_q^-$. Therefore, the gap condition is satisfied.

## D.2  TOKEN-LEVEL ANALYSIS LOWER BOUND (PROOF OF THEOREM 4)

Under the token-level formulation of (1), recall that $T_\infty$ denotes the maximum length of a sequence $\vec{\boldsymbol{o}} \in \mathcal{O}$. We then aim to show for any $T_\infty$, $L_{\mathrm{p}}$, and $G_{\mathrm{p}}$ with $G_{\mathrm{p}} \ge L_{\mathrm{p}}^{1/2}$, there exists a problem instance where we have $\Delta\mu_q(k) \ge \Delta\mu_q > 0$ and for $\eta$ satisfying

$$
\frac{120\Delta\mu_q}{T_\infty^{1/2} \cdot L_{\mathrm{p}} + T_\infty \cdot G_{\mathrm{p}}^2} \le \eta \le \frac{1}{2\sqrt{T_\infty^{1/2} \cdot L_{\mathrm{p}} + T_\infty G_{\mathrm{p}}^2}}\,,
$$

with $0 < \Delta\mu_q \le \frac{1}{240}\sqrt{T_\infty^{1/2} \cdot L_{\mathrm{p}} + T_\infty \cdot G_{\mathrm{p}}^2}$, we'll have degrading performance:

$$
J_q(k) < J_q(k - 1) \quad \text{and} \quad \lim_{K \to \infty} J_q(K) = 0\,.
$$

Similar to the proof of Theorem 2, we'll use a token space of $\{o_1, o_{-1}, o_{-2}\}$ and we'll use a linear feature representation for the logit $\boldsymbol{h}_\theta(q, \vec{\boldsymbol{o}}_{\le t}) := \langle \boldsymbol{\phi}(q, \vec{\boldsymbol{o}}_t), \theta \rangle$ at each layer $t \in [T_\infty]$. Note that the feature map only depends on the last token in the subsequence $\vec{\boldsymbol{o}}_{\le t}$ and is the same for each layer $t$. Each feature map will be

$$
\boldsymbol{\phi}(q, o_1) = \frac{G_{\mathrm{p}}}{2}\begin{pmatrix} 0 \\ 1 \end{pmatrix}, \qquad \boldsymbol{\phi}(q, o_{-1}) = \frac{G_{\mathrm{p}}}{2}\begin{pmatrix} 1 \\ 0 \end{pmatrix}, \qquad \boldsymbol{\phi}(q, o_{-2}) = \frac{G_{\mathrm{p}}}{2}\begin{pmatrix} -1 \\ 0 \end{pmatrix}.
$$

The optimization scheme will be the same as in the sequence-level analysis:

**Initialization:**

$$
\theta_0 := \frac{\eta}{3}\begin{pmatrix} -1 \\ 0 \end{pmatrix}. \tag{40}
$$

**Step size:**  fixed $\eta_k = \eta$ satisfying

$$
\frac{120\,\Delta\mu_q}{T_\infty^{1/2} \cdot L_{\mathrm{p}} + T_\infty \cdot G_{\mathrm{p}}^2} \le \eta \le \frac{1}{2\sqrt{T_\infty^{1/2} \cdot L_{\mathrm{p}} + T_\infty \cdot G_{\mathrm{p}}^2}}. \tag{41}
$$

**Update direction:**

$$
\boldsymbol{w}_k = \frac{1}{3}\begin{pmatrix} (-1)^k \cdot 2 \\ -\delta \end{pmatrix} \qquad \delta := \frac{1}{10}\, \eta\, G_{\mathrm{p}}. \tag{42}
$$

with resulting parameter trajectory:

$$
\theta_k := \frac{\eta}{3}\begin{pmatrix} -(-1)^t \\ -\delta k \end{pmatrix}. \tag{43}
$$

We can confirm $\|\boldsymbol{w}_k\|_2 \le 1$ since $\eta\, G_{\mathrm{p}} \le 500$.

Now, let the positive space of responses consist of a single sequence using all token $o_1$'s, or $\mathcal{O}^+ := \{o_1^{T_\infty}\}$ so that $\mathcal{O}^- := \mathcal{O}\backslash\mathcal{O}^+$. The reward $r^\star(q, \vec{\boldsymbol{o}})$ will be 0 for negative responses and 1 for positive responses.

**Decreasing value:** Since at each layer $t \in [T_\infty]$, the policy chooses a token independent from the previous tokens, the value function is

$$J_q(k) = \pi_{\theta_k}(o_1^{T_\infty} \mid q) = \left( \frac{1}{1 + \left\{ \exp(\eta\,G_{\mathrm{p}}/6) + \exp(-\eta\,G_{\mathrm{p}}/6) \right\} \cdot \exp\left\{ (\delta\eta\,G_{\mathrm{p}}/6) \cdot k \right\}} \right)^{T_\infty},$$

which is decreasing with increasing $k$ and goes to $0$.

**Checking positive gap.** Consider even indices $t$ (the odd case is symmetric). Then

$$\boldsymbol{w}_k = \frac{1}{3} \begin{pmatrix} 2 \\ -\delta \end{pmatrix}. \tag{44}$$

We first claim that

$$\left\{ \mathbb{E}_{\vec{\boldsymbol{o}} \sim \pi_{\theta_k}(\cdot \mid q, \mathcal{O}^+)} - \mathbb{E}_{\vec{\boldsymbol{o}} \sim \pi_{\theta_k}(\cdot \mid q, \mathcal{O}^-)} \right\} \left[ \nabla_\theta \, \log \pi_{\theta_k}(\vec{\boldsymbol{o}} \mid q) \right]$$

$$= \left\{ \mathbb{E}_{\vec{\boldsymbol{o}} \sim \pi_{\theta_k}(\cdot \mid q, \mathcal{O}^+)} - \mathbb{E}_{\vec{\boldsymbol{o}} \sim \pi_{\theta_k}(\cdot \mid q, \mathcal{O}^-)} \right\} \left[ \sum_{t=1}^{T_\infty} \nabla_\theta \, \log(\pi_{\theta_k}(\vec{\boldsymbol{o}}_t \mid q, \vec{\boldsymbol{o}}_{<t})) \right]$$

$$= \left\{ \mathbb{E}_{\vec{\boldsymbol{o}} \sim \pi_{\theta_k}(\cdot \mid q, \mathcal{O}^+)} - \mathbb{E}_{\vec{\boldsymbol{o}} \sim \pi_{\theta_k}(\cdot \mid q, \mathcal{O}^-)} \right\} \left[ \sum_{t=1}^{T_\infty} \phi(q, \vec{\boldsymbol{o}}_t) \right].$$

The last equality above is true because the normalizing constants of each $\pi_\theta(\vec{\boldsymbol{o}}_t \mid q, \vec{\boldsymbol{o}}_{<t})$ only do not depend on $\vec{\boldsymbol{o}}_{\leq t}$. Thus, the gradient log of the normalizing constants will cancel out from the positive and negative expectations.

We'll next simplify notation to avoid normalizing constants and let $\boldsymbol{g}_q^+ := \mathbb{E}_{\vec{\boldsymbol{o}} \sim \pi_{\theta_k}(\cdot \mid q, \mathcal{O}^+)} \left[ \sum_{t=1}^{T_\infty} \phi(q, \vec{\boldsymbol{o}}_t) \right]$ and define $\boldsymbol{g}_q^-$ analogously.

We next note

$$\boldsymbol{g}_q^+ = \mathbb{E}_{\vec{\boldsymbol{o}} \sim \pi_{\theta_k}(\cdot \mid q, \mathcal{O}^+)} \left[ \sum_{t=1}^{T_\infty} \phi(q, \vec{\boldsymbol{o}}_t) \right] = \frac{T_\infty \cdot G_{\mathrm{p}}}{2} \begin{pmatrix} 0 \\ 1 \end{pmatrix}.$$

Next, the negative term is

$$\boldsymbol{g}_q^- = \mathbb{E}_{\vec{\boldsymbol{o}} \sim \pi_{\theta_k}(\cdot \mid q, \mathcal{O}^-)} \left[ \sum_{t=1}^{T_\infty} \phi(q, \vec{\boldsymbol{o}}_t) \right]$$

$$= \mathbb{E}_{\vec{\boldsymbol{o}} \sim \pi_{\theta_k}(\cdot \mid q, \mathcal{O}^-)} \left[ \frac{G_{\mathrm{p}}}{2} \cdot \left( \begin{pmatrix} 1 \\ 0 \end{pmatrix} \cdot (N(\vec{\boldsymbol{o}}, -1) - N(\vec{\boldsymbol{o}}, -2)) + \begin{pmatrix} 0 \\ 1 \end{pmatrix} \cdot N(\vec{\boldsymbol{o}}, 1) \right） \right],$$

where $N(\vec{\boldsymbol{o}}, j) := \sum_{t=1}^{T_\infty} \mathbb{1}\{\vec{\boldsymbol{o}}_t = o_j\}$ is the count of tokens which are equal to $o_j$.

Now, acknowledging that $\pi_{\theta_k}(\vec{\boldsymbol{o}} \mid q, \mathcal{O}^-) = \frac{\pi_{\theta_k}(\vec{\boldsymbol{o}} \mid q) \cdot \mathbb{1}\{\vec{\boldsymbol{o}} \in \mathcal{O}_q^-\}}{1 - J_q(k)}$, we have

$$\mathbb{E}_{\vec{\boldsymbol{o}} \sim \pi_{\theta_k}^-(\cdot \mid q)}[N(\vec{\boldsymbol{o}}, -1)] = \frac{T_\infty}{1 - J_q(k)} \cdot \frac{\exp(-\eta G_{\mathrm{p}}/6)}{\exp(-\eta G_{\mathrm{p}}/6) + \exp(\eta G_{\mathrm{p}}/6) + \exp(-(\delta\eta G_{\mathrm{p}}/6) \cdot k)}$$

$$\mathbb{E}_{\vec{\boldsymbol{o}} \sim \pi_{\theta_k}^-(\cdot \mid q)}[N(\vec{\boldsymbol{o}}, -2)] = \frac{T_\infty}{1 - J_q(k)} \cdot \frac{\exp(\eta G_{\mathrm{p}}/6)}{\exp(-\eta G_{\mathrm{p}}/6) + \exp(\eta G_{\mathrm{p}}/6) + \exp(-(\delta\eta G_{\mathrm{p}}/6) \cdot k)}$$

$$\mathbb{E}_{\vec{\boldsymbol{o}} \sim \pi_{\theta_k}^-(\cdot \mid q)}[N(\vec{\boldsymbol{o}}, 1)] \leq \frac{T_\infty}{1 - J_q(k)} \cdot \frac{1}{1 + \left\{ \exp(\eta\,G_{\mathrm{p}}/6) + \exp(-\eta\,G_{\mathrm{p}}/6) \right\} \cdot \exp\left\{ (\delta\eta\,G_{\mathrm{p}}/6) \cdot k \right\}},$$

where the last inequality follows from bounding $\mathbb{1}\{\vec{\boldsymbol{o}}_t = o_1\} \cdot \mathbb{1}\{\vec{\boldsymbol{o}} \in \mathcal{O}_q^-\} \leq \mathbb{1}\{\vec{\boldsymbol{o}}_t = o_1\}$.

Thus, we have:

$$-\langle \boldsymbol{w}_k, \boldsymbol{g}_q^- \rangle \geq \frac{G_{\mathrm{p}} T_\infty}{3(1 - J_q(k))} \cdot \frac{\exp(\eta G_{\mathrm{p}}/3) - 1}{1 + \exp(\eta G_{\mathrm{p}}/3) + \exp(\eta G_{\mathrm{p}}/3 - (\delta\eta G_{\mathrm{p}}/6) \cdot k)}$$

$$+ \frac{\delta\,G}{6} \cdot \left( \frac{T_\infty}{1 - J_q(k)} \right) \cdot \frac{1}{1 + \left\{ \exp(\eta\,G_{\mathrm{p}}/6) + \exp(-\eta\,G_{\mathrm{p}}/6) \right\} \cdot \exp\left\{ (\delta\eta\,G_{\mathrm{p}}/6) \cdot k \right\}}.$$

Next, we note the elementary inequality $\frac{e^x - 1}{1 + 2e^x} \geq 0.3x$ for $x \in [0, 0.5]$. Then, since $J_q(k) \neq 0$ for all $t$, we have

$$\langle \boldsymbol{w}_k, \boldsymbol{g}_q^+ - \boldsymbol{g}_q^- \rangle \geq -\frac{T_\infty \cdot G_{\mathrm{p}} \cdot \delta}{6} + \frac{0.3 G_{\mathrm{p}}^2 \cdot T_\infty \cdot \eta}{9}$$

$$= T_\infty \cdot G_{\mathrm{p}} \cdot \left( -\frac{\delta}{6} + \frac{0.3}{9} \cdot \eta \cdot G_{\mathrm{p}} \right)$$

$$\geq \frac{T_\infty \cdot \eta \cdot G_{\mathrm{p}}^2}{60}$$

$$\geq \Delta \mu_q \,,$$

where the last inequality is true by $\eta \, T_\infty \, (G_{\mathrm{p}}^2 + L_{\mathrm{p}}) \geq \Delta \mu_q \cdot 120$ and $L_{\mathrm{p}} \leq G_{\mathrm{p}}^2$.

# E  FIRST-ORDER ANALYSIS OF THE LOG-ODDS CHANGE

In this section, we analyze how the objective value's log-odds ratio changes over a single update step. To simplify the notation, let us consider the state at a single update step $k$. We will use the following shorthand: step size $\eta := \eta_k$, update direction $\boldsymbol{w} = \boldsymbol{w}_k$, parameters $\theta_{\mathrm{old}} := \theta_k$ and $\theta := \theta_{k+1}$, objective values $J_q^{\mathrm{old}} := J_q(k)$ and $J_q := J_q(k+1)$. Our goal is to prove the following relationship:

$$\log \left( \frac{J_q}{1 - J_q} \right) - \log \left( \frac{J_q^{\mathrm{old}}}{1 - J_q^{\mathrm{old}}} \right) \;=\; \Delta \mu_q (\pi_{\theta_{\mathrm{old}}}) \, \eta + \mathcal{O}(\eta^2) \,,$$

where the gap alignment $\Delta \mu_q$ is defined in equation (11). Essentially, this equation shows a first-order Taylor expansion of the change in the log-odds ratio. The key insight is that the term $\Delta \mu_q$ emerges as the linear coefficient for the step size $\eta$.

We will prove this fundamental relationship at two distinct levels of granularity:

**Trajectory Level:** The analysis for entire generation trajectories is presented in Lemma 1, with the full proof in Section E.1.

**Token Level:** The corresponding result for token-level consideration is established in Lemma 2, with its proof located in Section E.2.

## E.1  PROOF OF LEMMA 1

### E.1.1  OVERVIEW

We first formulate Lemma 3 below, which establishes an equivalent expression for the difference of logarithms. The proof is deferred to Section E.1.2.

**Lemma 3.** *For policies $\pi_{\theta_{\mathrm{old}}}$ and $\pi_\theta$, it holds that*

$$\log \left( \frac{J_q}{1 - J_q} \right) - \log \left( \frac{J_q^{\mathrm{old}}}{1 - J_q^{\mathrm{old}}} \right)$$

$$= \log \mathbb{E}_{\vec{\boldsymbol{o}} \sim \pi_{\theta_{\mathrm{old}}}(\cdot \,|\, q, \, \mathcal{O}_q^+)} \left[ \exp \left\{ (\log \pi_\theta - \log \pi_{\theta_{\mathrm{old}}})(\vec{\boldsymbol{o}} \,|\, q) \right\} \right]$$

$$- \log \mathbb{E}_{\vec{\boldsymbol{o}} \sim \pi_{\theta_{\mathrm{old}}}(\cdot \,|\, q, \, \mathcal{O}_q^-)} \left[ \exp \left\{ (\log \pi_\theta - \log \pi_{\theta_{\mathrm{old}}})(\vec{\boldsymbol{o}} \,|\, q) \right\} \right]. \quad (45)$$

The difference of logarithms from Lemma 3 is central to our analysis of the optimization scheme. As shown in equation (45), we express this term as the difference between two log moment-generating functions (MGFs). These MGFs are derived from the conditional probabilities over the correct (positive) and incorrect (negative) solution spaces, $\mathcal{O}_q^+$ and $\mathcal{O}_q^-$, respectively.

In Lemma 4 below, we approximate the log MGFs based on the expectations of the random variables. See Section E.1.3 for the proof.

**Lemma 4.** *For any random variable $X$, we have:*

$$\left| \log \mathbb{E}[e^X] - \mathbb{E}[X] \right| \;\leq\; 2 \, \|X\|_\infty^2 \,. \quad (46)$$

We now combine Lemmas 3 and 4 to prove Lemma 1. In what follows, we introduce a shorthand $\boldsymbol{L}_\theta := \log \pi_\theta$. In our analysis recall that $\theta - \theta_{\text{old}} = \eta_k \cdot \boldsymbol{w}_k$ with $\|\boldsymbol{w}_k\|_2 \leq 1$. We denote $\eta := \eta_k$ and $\boldsymbol{w} := \boldsymbol{w}_k$ for short.

We apply the bound from equation (46) in Lemma 4 to the random variable $(\boldsymbol{L}_\theta - \boldsymbol{L}_{\theta_{\text{old}}})(q, \vec{\boldsymbol{o}})$. By evaluating this for the distributions corresponding to the positive space, $\vec{\boldsymbol{o}} \sim \pi_{\theta_{\text{old}}}(\cdot \mid q, \mathcal{O}_q^+)$, and the negative space, $\vec{\boldsymbol{o}} \sim \pi_{\theta_{\text{old}}}(\cdot \mid q, \mathcal{O}_q^-)$, we find that

$$\left| \log \left( \frac{J_q}{1 - J_q} \right) - \log \left( \frac{J_q^{\text{old}}}{1 - J_q^{\text{old}}} \right) - T_1 \right| \leq 4 \left\| \boldsymbol{L}_\theta - \boldsymbol{L}_{\theta_{\text{old}}} \right\|_\infty^2, \tag{47}$$

where

$$T_1 := \left\{ \mathbb{E}_{\vec{\boldsymbol{o}} \sim \pi_{\theta_{\text{old}}}(\cdot \mid q, \mathcal{O}_q^+)} - \mathbb{E}_{\vec{\boldsymbol{o}} \sim \pi_{\theta_{\text{old}}}(\cdot \mid q, \mathcal{O}_q^-)} \right\} \left[ (\boldsymbol{L}_\theta - \boldsymbol{L}_{\theta_{\text{old}}})(q, \vec{\boldsymbol{o}}) \right].$$

To establish the final bound in Lemma 1, we need to prove two intermediate results:

1. $T_1 = \Delta \mu_q(\pi_{\theta_{\text{old}}}) \cdot \eta + \mathcal{O}(\eta^2)$
2. $\|\boldsymbol{L}_\theta - \boldsymbol{L}_{\theta_{\text{old}}}\|_\infty = \mathcal{O}(\eta)$.

Once we show these two relationships hold, the desired bound (28) follows directly by substituting them into inequality (47).

**Analyzing term $T_1$:** For term $T_1$, Assumption 1 on the smoothness of function $\boldsymbol{L}_\theta$ implies that

$$\left| (\boldsymbol{L}_\theta - \boldsymbol{L}_{\theta_{\text{old}}})(q, \vec{\boldsymbol{o}}) - \langle \nabla_\theta \boldsymbol{L}_{\theta_{\text{old}}}(q, \vec{\boldsymbol{o}}), \theta - \theta_{\text{old}} \rangle \right| \leq \frac{L_{\text{o}}}{2} \|\theta - \theta_{\text{old}}\|_2^2.$$

According to our optimization scheme $\theta = \theta_{\text{old}} + \eta \cdot \boldsymbol{w}$, it follows that

$$\left| (\boldsymbol{L}_\theta - \boldsymbol{L}_{\theta_{\text{old}}})(q, \vec{\boldsymbol{o}}) - \eta \left\{ \boldsymbol{w} \cdot \nabla_\theta \boldsymbol{L}_{\theta_{\text{old}}}(q, \vec{\boldsymbol{o}}) \right\} \right| \leq \frac{L_{\text{o}}}{2} \eta^2. \tag{48}$$

Recalling the definition (8) of vectors $\boldsymbol{g}_q^+ = \boldsymbol{g}_q^+(\pi_{\theta_{\text{old}}})$ and $\boldsymbol{g}_q^- = \boldsymbol{g}_q^-(\pi_{\theta_{\text{old}}})$, we find that

$$\boldsymbol{g}_q^+ - \boldsymbol{g}_q^- = \left\{ \mathbb{E}_{\pi_{\theta_{\text{old}}}^+(\cdot|q)} - \mathbb{E}_{\pi_{\theta_{\text{old}}}^-(\cdot|q)} \right\} \left[ \nabla_\theta \boldsymbol{L}_{\theta_{\text{old}}}(q, \vec{\boldsymbol{o}}) \right].$$

Moreover, the definition (11) of gap $\Delta \mu_q$ leads to

$$\Delta \mu_q(\pi_{\theta_{\text{old}}}) = \langle \boldsymbol{w}, \boldsymbol{g}_q^+ - \boldsymbol{g}_q^- \rangle = \left\{ \mathbb{E}_{\pi_{\theta_{\text{old}}}^+(\cdot|q)} - \mathbb{E}_{\pi_{\theta_{\text{old}}}^-(\cdot|q)} \right\} \left[ \boldsymbol{w} \cdot \nabla_\theta \boldsymbol{L}_{\theta_{\text{old}}}(q, \vec{\boldsymbol{o}}) \right].$$

Therefore, taking expectations $\left\{ \mathbb{E}_{\pi_{\theta_{\text{old}}}^+(\cdot|q)} - \mathbb{E}_{\vec{\boldsymbol{o}} \sim \pi_{\theta_{\text{old}}}^-(\cdot|q)} \right\}(\cdot)$ of the terms inside the absolute value of equation (48), we get

$$\left| T_1 - \Delta \mu_q(\pi_{\theta_{\text{old}}}) \eta \right| \leq \frac{L_{\text{o}}}{2} \eta^2. \tag{49}$$

**Bounding norm $\|\boldsymbol{L}_\theta - \boldsymbol{L}_{\theta_{\text{old}}}\|_\infty$:** From inequality (48), we also have

$$\|\boldsymbol{L}_\theta - \boldsymbol{L}_{\theta_{\text{old}}}\|_\infty \leq \eta \sup_{\vec{\boldsymbol{o}} \in \mathcal{O}} \left\{ \boldsymbol{w} \cdot \nabla_\theta \boldsymbol{L}_{\theta_{\text{old}}}(q, \vec{\boldsymbol{o}}) \right\} + \frac{L_{\text{o}}}{2} \eta^2$$

$$\leq \eta \sup_{\vec{\boldsymbol{o}} \in \mathcal{O}} \|\nabla_\theta \boldsymbol{L}_{\theta_{\text{old}}}(q, \vec{\boldsymbol{o}})\|_2 + \frac{L_{\text{o}}}{2} \eta^2 \leq G_{\text{o}} \eta + \frac{L_{\text{o}}}{2} \eta^2. \tag{50}$$

**Finalizing the proof:** Combining our previous bounds (47), (49) and (50), we get

$$\left| \log \left( \frac{J_q}{1 - J_q} \right) - \log \left( \frac{J_q^{\text{old}}}{1 - J_q^{\text{old}}} \right) - \Delta \mu_q(\pi_{\theta_{\text{old}}}) \eta \right|$$

$$\leq 4 \|\boldsymbol{L}_\theta - \boldsymbol{L}_{\theta_{\text{old}}}\|_\infty^2 + \left| T_1 - \Delta \mu_q(\pi_{\theta_{\text{old}}}) \eta \right|$$

$$\leq \left\{ (2 G_{\text{o}} + L_{\text{o}} \eta)^2 + \frac{L_{\text{o}}}{2} \right\} \eta^2 \leq (L_{\text{o}} + 8 G_{\text{o}}^2) \eta^2,$$

where the last inequality follows from the condition $\eta \leq \frac{1}{2\sqrt{L_{\text{o}}}}$. This establishes inequality (28) and completes the proof of Lemma 1.

### E.1.2 PROOF OF LEMMA 3

Let $\boldsymbol{L}_\theta(\vec{\boldsymbol{o}}, q) := \log \pi_\theta(\vec{\boldsymbol{o}} \mid q)$ be the log-likelihood function and define $\boldsymbol{L}_{\theta_{\text{old}}}$ analogously. The updated policy $\pi_\theta$ can be expressed in terms of the old policy $\pi_{\theta_{\text{old}}}$ as follows:

$$\pi_\theta(\vec{\boldsymbol{o}} \mid q) \propto \pi_{\theta_{\text{old}}}(\vec{\boldsymbol{o}} \mid q) \exp\left\{ (\boldsymbol{L}_\theta - \boldsymbol{L}_{\theta_{\text{old}}})(\vec{\boldsymbol{o}} \mid q) \right\}.$$

Therefore, we have

$$\frac{J_q}{1 - J_q} = \frac{\sum_{\vec{\boldsymbol{o}} \in \mathcal{O}_q^+} \pi_\theta(\vec{\boldsymbol{o}} \mid q)}{\sum_{\vec{\boldsymbol{o}} \in \mathcal{O}_q^-} \pi_\theta(\vec{\boldsymbol{o}} \mid q)} = \frac{\sum_{\vec{\boldsymbol{o}} \in \mathcal{O}_q^+} \pi_{\theta_{\text{old}}}(\vec{\boldsymbol{o}} \mid q) \exp\left\{ (\boldsymbol{L}_\theta - \boldsymbol{L}_{\theta_{\text{old}}})(q, \vec{\boldsymbol{o}}) \right\}}{\sum_{\vec{\boldsymbol{o}} \in \mathcal{O}_q^-} \pi_{\theta_{\text{old}}}(\vec{\boldsymbol{o}} \mid q) \exp\left\{ (\boldsymbol{L}_\theta - \boldsymbol{L}_{\theta_{\text{old}}})(q, \vec{\boldsymbol{o}}) \right\}} =: \frac{T_+}{T_-}. \tag{51}$$

Regarding the numerator $T_+$, we use the conditional probability relationship

$$\pi_{\theta_{\text{old}}}(\vec{\boldsymbol{o}} \mid q) = J_q^{\text{old}} \, \pi_{\theta_{\text{old}}}(\vec{\boldsymbol{o}} \mid q, \mathcal{O}_q^+) = J_q^{\text{old}} \, \pi_{\theta_{\text{old}}}^+(\vec{\boldsymbol{o}} \mid q) \qquad \text{for } \vec{\boldsymbol{o}} \in \mathcal{O}_q^+$$

and find that

$$T_+ = J_q^{\text{old}} \sum_{\vec{\boldsymbol{o}} \in \mathcal{O}_q^+} \pi_{\theta_{\text{old}}}^+(\vec{\boldsymbol{o}} \mid q) \exp\left\{ (\boldsymbol{L}_\theta - \boldsymbol{L}_{\theta_{\text{old}}})(q, \vec{\boldsymbol{o}}) \right\}$$

$$= J_q^{\text{old}} \cdot \mathbb{E}_{\pi_{\theta_{\text{old}}}^+(\cdot \mid q)}\left[ \exp\left\{ (\boldsymbol{L}_\theta - \boldsymbol{L}_{\theta_{\text{old}}})(q, \vec{\boldsymbol{o}}) \right\} \right]. \tag{52a}$$

Similarly, for the denominator $T_-$, we apply equality

$$\pi_{\theta_{\text{old}}}(\vec{\boldsymbol{o}} \mid q) = (1 - J_q^{\text{old}}) \, \pi_{\theta_{\text{old}}}(\vec{\boldsymbol{o}} \mid q, \mathcal{O}_q^-) = (1 - J_q^{\text{old}}) \, \pi_{\theta_{\text{old}}}^-(\vec{\boldsymbol{o}} \mid q) \qquad \text{for } \vec{\boldsymbol{o}} \in \mathcal{O}_q^-.$$

It follows that

$$T_- = (1 - J_q^{\text{old}}) \cdot \mathbb{E}_{\pi_{\theta_{\text{old}}}^-(\cdot \mid q)}\left[ \exp\left\{ (\boldsymbol{L}_\theta - \boldsymbol{L}_{\theta_{\text{old}}})(q, \vec{\boldsymbol{o}}) \right\} \right]. \tag{52b}$$

Then, substituting equations (52a) and (52b) into equation (51), we obtain precisely equation (45) stated in Lemma 3.

### E.1.3 PROOF OF LEMMA 4

We start by centering the random variable $X$. Let us define $Y := X - \mathbb{E}[X]$, which gives us a new variable with a mean of zero. This simplifies the core expression:

$$\log \mathbb{E}[e^X] - \mathbb{E}[X] = \log \mathbb{E}[e^{\mathbb{E}[X]+Y}] - \mathbb{E}[X] = \log \mathbb{E}[e^Y].$$

By Jensen's inequality, the term $\log \mathbb{E}[e^Y]$ is always non-negative. This allows us to safely drop the absolute value bars:

$$\left| \log \mathbb{E}[e^X] - \mathbb{E}[X] \right| = \log \mathbb{E}[e^Y].$$

Finally, applying Hoeffding's lemma to this resulting log moment-generating function bounds it by at most $2\|X\|_\infty^2$. This establishes the result in Lemma 4.

### E.2 PROOF OF LEMMA 2

### E.2.1 OVERVIEW

Our proof proceeds in two main steps. First, we establish a lower bound on the improvement in the log-odds of value $J_q$:

$$\log\left( \frac{J_q}{1 - J_q} \right) - \log\left( \frac{J_q^{\text{old}}}{1 - J_q^{\text{old}}} \right)$$

$$\geq \left\{ \log\left( \mathbb{E}^+[\exp(X)] \right) - \log\left( \mathbb{E}^-[\exp(X)] \right) \right\} - L_{\text{p}} T_\infty \cdot \eta^2, \tag{53}$$

where the random variable $X$ is defined as

$$X := \eta \left\{ \boldsymbol{w} \cdot \nabla_\theta \log \pi_{\theta_{\mathrm{old}}}(\vec{\boldsymbol{o}} \mid q) \right\} = \left\langle \nabla_\theta \log \pi_{\theta_{\mathrm{old}}}(\vec{\boldsymbol{o}} \mid q), \theta - \theta_{\mathrm{old}} \right\rangle. \qquad (54)$$

Here $\mathbb{E}^+$ and $\mathbb{E}^-$ denote expectations over the conditional distributions of positive and negative responses, i.e., $\mathbb{E}_{\vec{\boldsymbol{o}} \sim \pi_{\theta_{\mathrm{old}}}(\cdot \mid q, \mathcal{O}_q^+)}$ and $\mathbb{E}_{\vec{\boldsymbol{o}} \sim \pi_{\theta_{\mathrm{old}}}(\cdot \mid q, \mathcal{O}_q^-)}$, respectively.

Second, we bound the difference between the log-moment-generating functions (log-MGFs) that appears on the right-hand side of inequality (53):

$$\log \left( \mathbb{E}^+[\exp(X)] \right) - \log \left( \mathbb{E}^-[\exp(X)] \right) \geq \Delta\mu_q(\pi_{\theta_{\mathrm{old}}}) \cdot \eta - \frac{G_{\mathrm{p}}^2 T_{\psi_1}}{J_q^{\mathrm{old}}} \cdot \eta^2, \qquad (55)$$

which holds under the condition that

$$\eta \leq \frac{1}{\sqrt{2\, G_{\mathrm{p}}^2 \cdot T_{\psi_1}}}.$$

Combining the bounds from (53) and (55) directly establishes the result in Lemma 2. We now prove these two intermediate claims in Sections E.2.2 and E.2.3.

### E.2.2 Proof of Inequality (53)

Our starting point is the smoothness property of the log-policy, as stated in Assumption 2. For any prompt-response pair $(q, \vec{\boldsymbol{o}})$, the function $\log \pi_\theta(\vec{\boldsymbol{o}} \mid q)$ is $(L_{\mathrm{p}} \cdot T_\infty)$-smooth with respect to $\theta$. A standard result for smooth functions is that:

$$\left| (\log \pi_\theta - \log \pi_{\theta_{\mathrm{old}}})(\vec{\boldsymbol{o}} \mid q) - \left\langle \nabla_\theta \log \pi_{\theta_{\mathrm{old}}}(\vec{\boldsymbol{o}} \mid q), \theta - \theta_{\mathrm{old}} \right\rangle \right| \leq \frac{L_{\mathrm{p}} \cdot T_\infty}{2} \|\theta - \theta_{\mathrm{old}}\|_2^2.$$

By substituting the definition of the random variable $X$ from equation (54) and using the gradient update rule $\theta - \theta_{\mathrm{old}} = \eta \cdot \boldsymbol{w}$ (where $\|\boldsymbol{w}\|_2 \leq 1$), this inequality simplifies to:

$$\left| (\log \pi_\theta - \log \pi_{\theta_{\mathrm{old}}})(\vec{\boldsymbol{o}} \mid q) - X \right| \leq \frac{L_{\mathrm{p}} T_\infty}{2} \cdot \eta^2.$$

This directly implies the following bounds on the conditional expectations:

$$\log \mathbb{E}_{\vec{\boldsymbol{o}} \sim \pi_{\theta_{\mathrm{old}}}(\cdot \mid q, \mathcal{O}_q^+)} \left[ \exp \left\{ (\log \pi_\theta - \log \pi_{\theta_{\mathrm{old}}})(\vec{\boldsymbol{o}} \mid q) \right\} \right] \geq \log \left( \mathbb{E}^+[\exp(X)] \right) - \frac{L_{\mathrm{p}} T_\infty}{2} \cdot \eta^2.$$

$$\log \mathbb{E}_{\vec{\boldsymbol{o}} \sim \pi_{\theta_{\mathrm{old}}}(\cdot \mid q, \mathcal{O}_q^-)} \left[ \exp \left\{ (\log \pi_\theta - \log \pi_{\theta_{\mathrm{old}}})(\vec{\boldsymbol{o}} \mid q) \right\} \right] \leq \log \left( \mathbb{E}^-[\exp(X)] \right) + \frac{L_{\mathrm{p}} T_\infty}{2} \cdot \eta^2.$$

Subtracting the second inequality from the first yields the desired bound (53).

### E.2.3 Proof of Inequality (55)

To prove inequality (55), we will establish and combine three intermediate results.

First, we relate the ratio of conditional MGFs to the unconditional MGF:

$$\log \left( \frac{\mathbb{E}^+[e^X]}{\mathbb{E}^-[e^X]} \right) \geq -\frac{1}{1 - J_q^{\mathrm{old}}} \log \mathbb{E}[e^X] + \frac{1}{1 - J_q^{\mathrm{old}}} \log \mathbb{E}^+[e^X]. \qquad (56)$$

Here $\mathbb{E}$ denotes the expectation over the entire response space $\mathcal{O} = \mathcal{O}_q^+ \cup \mathcal{O}_q^-$.

Next, we bound the two terms on the right-hand side of inequality (56) separately. We show that

$$\frac{1}{1 - J_q^{\mathrm{old}}} \log \mathbb{E}^+[e^X] \geq \Delta\mu_q(\pi_{\theta_{\mathrm{old}}}) \cdot \eta, \qquad \text{and} \qquad (57\mathrm{a})$$

$$\frac{1}{1 - J_q^{\mathrm{old}}} \log \mathbb{E}[e^X] \leq \frac{G_{\mathrm{p}}^2 \cdot T_{\psi_1}}{1 - J_q^{\mathrm{old}}} \cdot \eta^2 \qquad \text{if } \eta \leq \frac{1}{\sqrt{2\, G_{\mathrm{p}}^2 \cdot T_{\psi_1}}}. \qquad (57\mathrm{b})$$

Combining these three bounds (56), (57a) and (57b) directly yields the target inequality (55).

We now prove each claim in turn.

**Proof of Inequality** (56): This result follows from a general property of logarithms derived from the weighted AM-GM inequality. For any real numbers $u, v > 0$ and any $p \in (0, 1)$, the weighted AM-GM inequality states $pu + (1 - p)v \geq u^p v^{1-p}$. Dividing by $v$ and taking the logarithm gives

$$\log(u/v) \leq \frac{1}{p} \left( \log(pu + (1 - p)v) - \log(v) \right).$$

We apply this inequality by setting $u := \mathbb{E}^-[e^X]$, $v := \mathbb{E}^+[e^X]$ and $p := 1 - J_q^{\text{old}}$. This substitution yields

$$\log\left( \frac{\mathbb{E}^-[e^X]}{\mathbb{E}^+[e^X]} \right) \leq \frac{1}{1 - J_q^{\text{old}}} \log \left\{ J_q^{\text{old}} \, \mathbb{E}^+[e^X] + (1 - J_q^{\text{old}}) \, \mathbb{E}^-[e^X] \right\} - \frac{1}{1 - J_q^{\text{old}}} \log \mathbb{E}^+[e^X]. \tag{58}$$

By the law of total expectation, the term in the curly braces is simply the unconditional expectation $\mathbb{E}[e^X]$:

$$\mathbb{E}[e^X] = J_q^{\text{old}} \, \mathbb{E}^+[e^X] + (1 - J_q^{\text{old}}) \, \mathbb{E}^-[e^X]. \tag{59}$$

Substituting (59) into (58) completes the proof of inequality (56).

**Proof of Inequality** (57a): By Jensen's inequality, $\log \mathbb{E}^+[e^X] \geq \mathbb{E}^+[X]$. Therefore, we have

$$\frac{\log \mathbb{E}^+[e^X]}{1 - J_q^{\text{old}}} \geq \frac{\mathbb{E}^+[X]}{1 - J_q^{\text{old}}},$$

The remainder of the proof is dedicated to showing that the right-hand side is exactly equal to the desired term:

$$\frac{\mathbb{E}^+[X]}{1 - J_q^{\text{old}}} = \Delta \mu_q(\pi_{\theta_{\text{old}}}) \cdot \eta. \tag{60}$$

To prove this, we start with the policy gradient identity from equation (10):

$$\boldsymbol{w} \cdot \nabla_\theta J_q(\pi_{\theta_{\text{old}}}) = J_q^{\text{old}}(1 - J_q^{\text{old}}) \cdot \Delta \mu_q(\pi_{\theta_{\text{old}}}).$$

We can express policy gradient $\nabla_\theta J_q(\pi_{\theta_{\text{old}}})$ in terms of an expectation over negative responses. Using the fact that $J_q(\pi_{\theta_{\text{old}}}) = \mathbb{P}_{\vec{\boldsymbol{o}} \sim \pi_{\theta_{\text{old}}}(\cdot|q)}[\vec{\boldsymbol{o}} \in \mathcal{O}_q^+]$, we get

$$\nabla_\theta J_q(\pi_{\theta_{\text{old}}}) = \nabla_\theta \mathbb{P}_{\vec{\boldsymbol{o}} \sim \pi_{\theta_{\text{old}}}(\cdot|q)}[\vec{\boldsymbol{o}} \in \mathcal{O}_q^+] = \mathbb{E}\left[ \mathbb{1}\{\vec{\boldsymbol{o}} \in \mathcal{O}_q^+\} \cdot \nabla_\theta \log \pi_{\theta_{\text{old}}}(\vec{\boldsymbol{o}} \mid q) \right]$$
$$= J_q^{\text{old}} \cdot \mathbb{E}^+\left[ \nabla_\theta \log \pi_{\theta_{\text{old}}}(\vec{\boldsymbol{o}} \mid q) \right].$$

Substituting this back into our gradient identity yields:

$$\boldsymbol{w} \cdot J_q^{\text{old}} \cdot \mathbb{E}^+\left[ \nabla_\theta \log \pi_{\theta_{\text{old}}}(\vec{\boldsymbol{o}} \mid q) \right] = J_q^{\text{old}}(1 - J_q^{\text{old}}) \cdot \Delta \mu_q(\pi_{\theta_{\text{old}}}).$$

Dividing by $J_q^{\text{old}}$ and multiplying by $\eta$, we find

$$\eta \cdot \mathbb{E}^+\left[ \boldsymbol{w} \cdot \nabla_\theta \log \pi_{\theta_{\text{old}}}(\vec{\boldsymbol{o}} \mid q) \right] = (1 - J_q^{\text{old}}) \cdot \Delta \mu_q(\pi_{\theta_{\text{old}}}) \cdot \eta.$$

From the definition of $X$ in (54), the left-hand side is $\mathbb{E}^+[X]$. This confirms equation (60) and completes the proof of inequality (57a).

**Proof of Inequality** (57b): The final step is to bound the unconditional log-MGF, $\log \mathbb{E}[e^X]$. Our strategy is to view $X$ as the final value of a martingale and apply a variant of Hoeffding's inequality.

*Martingale Formulation.* Consider a random response $\vec{\boldsymbol{o}}$ generated from policy $\pi_{\theta_{\text{old}}}$, i.e., $\vec{\boldsymbol{o}} \sim \pi_{\theta_{\text{old}}}(\cdot \mid q)$. The random variable $X$ is a sum over the tokens in the response $\vec{\boldsymbol{o}}$:

$$X = \sum_{t=1}^{|\vec{\boldsymbol{o}}|} \eta \left\{ \boldsymbol{w} \cdot \nabla_\theta \log \pi_{\theta_{\text{old}}}(o_t \mid q, \vec{\boldsymbol{o}}_{<t}) \right\}.$$

Let $\mathcal{F}_t$ be the $\sigma$-field generated by the prompt $q$ and the response prefix $\vec{o}_{\leq t}$. We define a martingale difference sequence $\{\xi_t\}$:

$$\xi_t := \eta \left\{ \boldsymbol{w} \cdot \nabla_\theta \log \pi_{\theta_{\text{old}}}(o_t \mid q, \vec{o}_{<t}) \right\}. \tag{61}$$

Then $X_t = \sum_{t'=1}^{t} \xi_{t'}$ is a martingale with respect to the filtration $\{\mathcal{F}_t\}$, and $X = X_{|\vec{o}|}$. By Assumption 2, the increments are bounded: $|\xi_t| \leq G_{\text{p}} \cdot \eta$.

*Supermartingale and Optional Stopping.* Using Hoeffding's lemma (Boucheron et al., 2013, Lemma 2.6) on the conditional expectation, we have

$$\mathbb{E}\left[ \exp(2\,\xi_t) \mid \mathcal{F}_{t-1} \right] \leq \exp\left\{ 2\,G_{\text{p}}^2 \eta^2 \right\}.$$

This allows us to construct a supermartingale $\{\mathbf{M}_t\}_{t=1}^{T_\infty}$:

$$\mathbf{M}_t := \exp\left\{ 2\,X_t - 2\,G_{\text{p}}^2 \eta^2 \cdot t \right\} \qquad \text{for } t = 1, 2, \ldots.$$

Since $|\vec{o}| \leq T_\infty$ is a bounded stopping time, Doob's optional stopping theorem applies, giving $\mathbb{E}[\mathbf{M}_{|\vec{o}|}] \leq \mathbb{E}[\mathbf{M}_0] = 1$. This means

$$\mathbb{E}\left[ \exp\left\{ 2\,X - 2\,G_{\text{p}}^2 \eta^2 \cdot |\vec{o}| \right\} \right] = \mathbb{E}\left[ \mathbf{M}_{|\vec{o}|} \right] \leq 1.$$

*Cauchy-Schwarz and $\psi_1$ Norm.* We now isolate $\mathbb{E}[e^X]$ using the Cauchy-Schwarz inequality:

$$\mathbb{E}[e^X] \leq \mathbb{E}\left[ \exp\left\{ 2\,X - 2\,G_{\text{p}}^2 \eta^2 \cdot |\vec{o}| \right\} \right]^{\frac{1}{2}} \mathbb{E}\left[ \exp\left\{ 2\,G_{\text{p}}^2 \eta^2 \cdot |\vec{o}| \right\} \right]^{\frac{1}{2}} \leq \mathbb{E}\left[ \exp\left\{ 2\,G_{\text{p}}^2 \eta^2 \cdot |\vec{o}| \right\} \right]^{\frac{1}{2}},$$

where the last step used our supermartingale bound. Taking the logarithm gives

$$\log \mathbb{E}[e^X] \leq \frac{1}{2} \log \mathbb{E}\left[ \exp\left\{ 2\,G_{\text{p}}^2 \eta^2 \cdot |\vec{o}| \right\} \right].$$

Finally, we bound the remaining term using the sub-exponential ($\psi_1$-Orlicz) norm of the response length, $T_{\psi_1}$ from Assumption 3. By definition,

$$\mathbb{E}\left[ \exp\{ |\vec{o}|/T_{\psi_1} \} \right] \leq 2.$$

Consider a function $\phi(\lambda) := \log \mathbb{E}[\exp(\lambda\,|\vec{o}|)]$, which is convex with $\phi(0) = 0$. Under condition $0 \leq 2\,G_{\text{p}}^2 \eta^2 \leq 1/T_{\psi_1}$, the convexity implies

$$\phi(2\,G_{\text{p}}^2 \eta^2) \leq 2\,G_{\text{p}}^2 \eta^2 T_{\psi_1} \cdot \phi(1/T_{\psi_1}) \leq 2\log 2 \cdot G_{\text{p}}^2 \eta^2 T_{\psi_1}.$$

It follows that

$$\log \mathbb{E}\left[ \exp\left\{ 2\,G_{\text{p}}^2 \eta^2 \cdot |\vec{o}| \right\} \right] = \phi(2\,G_{\text{p}}^2 \eta^2) \leq 2\,G_{\text{p}}^2 \eta^2 \cdot T_{\psi_1}.$$

Plugging this back into our bound for $\log \mathbb{E}[e^X]$ gives

$$\log \mathbb{E}[e^X] \leq G_{\text{p}}^2 \eta^2 \cdot T_{\psi_1}.$$

This establishes inequality (57b) and completes the proof.

# F  FURTHER RESULT ON CONVERGENCE OF REWARD VS. DIVERGING CUMULATIVE GAP ALIGNMENT

Here, we show a stronger form of Theorem 1(b) holds, in that convergence $\lim_{K\to\infty} J_q(K) = 1$ implies divergence of the cumulative gap alignment $\lim_{K\to\infty} M(K) = +\infty$, establishing a converse to Theorem 1(b). We present the result for the trajectory-level formulation of Section 3 and the result for the token-level formulation of Section 4 is analogous.

Furthermore, the sum of policy gradient norms $\sum_{k=0}^{K-1} \|\nabla_\theta J_q(\pi_{\theta_k})\|_2$ may not correspond to such a

**Theorem 5.** *Consider a case where $J_q(0) < 1$. Assume the step sizes $\eta_k$ satisfy*

$$\eta_k \ \leq \ \frac{[\Delta\mu_q(k)]_+}{2\,(L_{\mathrm{o}} + 8\,G_{\mathrm{o}}^2)} \wedge \frac{1}{2\sqrt{L_{\mathrm{o}}}} \quad \textit{where } [\,\cdot\,]_+ = \max(0, \cdot)\,. \tag{62}$$

*(a) (**Convergence of Reward Implies Divergence of Cumulative Gap Alignment**) If $\lim\limits_{K\to\infty} J_q(K) = 1$, then $\lim\limits_{K\to\infty} M(K) = +\infty$.*

*(b) (**Vanishing Gradient Norms under Uniform Gap**) We have:*

$$\sum_{k=0}^{\infty}\|\nabla_\theta\, J_q(\pi_{\theta_k})\|_2^2 < 2L_{\mathrm{o}} \sum_{k=0}^{\infty} \frac{1 - J_q(0)}{1 - J_q(0) + J_q(0)\cdot \exp\left\{\frac{M(k)}{2}\right\}}\,.$$

*Furthermore, if $J_q(0) \in (0,1)$ and every update direction $\boldsymbol{w}_k$ provides a uniform gap, $\Delta\mu_q(k) \geq \Delta\mu_q > 0$ for all $k \geq 0$, a fixed step size $\eta \equiv \eta_k$ satisfying (62) yields $\sum_{k=0}^{\infty}\|\nabla_\theta\, J_q(\pi_{\theta_k})\|_2^2 < \infty$.*

*Proof.* We first show part (a). From Lemma 1, we have

$$\log\left(\frac{J_q(K)}{1 - J_q(K)}\right) - \log\left(\frac{J_q(0)}{1 - J_q(0)}\right) \ \leq \ \sum_{k=0}^{K-1} \Delta\mu_q(k)\,\eta_k + (L_{\mathrm{o}} + 8\,G_{\mathrm{o}}^2)\sum_{k=0}^{K-1}\eta_k^2$$

$$\leq \ \frac{3}{2}\sum_{k=0}^{K-1}[\Delta\mu_q(k)]_+\,\eta_k$$

$$= \ \frac{3}{2}\,M(K)\,.$$

Thus, if $J_q(K) \to 1$, we have $M(K) \to \infty$.

Next, we show part (b). From (10), and boundedness of the policy score (Assumption 1) we have

$$\|\nabla_\theta\, J_q(\pi_{\theta_k})\|_2 = J_q(\pi_{\theta_k})\cdot\{1 - J_q(\pi_{\theta_k})\}\cdot\|\boldsymbol{g}_q^+ - \boldsymbol{g}_q^-\|_2 \leq 2G_{\mathrm{o}}\cdot(1 - J_q(\pi_{\theta_k}))\,.$$

At the same time, from (30) and following the calculations of the proof of Section C.1.2, we have that

$$1 - J_q(k) \leq \frac{1 - J_q(0)}{1 - J_q(0) + J_q(0)\cdot \exp\left\{\frac{M(k)}{2}\right\}}\,.$$

Thus, $\|\nabla_\theta\, J_q(\pi_{\theta_k})\|_2 = \mathcal{O}\left(\exp\left\{-K\cdot\Delta\mu_q\cdot\eta/2\right\}\right)$ meaning $\sum_{k=0}^{\infty}\|\nabla_\theta\, J_q(\pi_{\theta_k})\|_2^2 < \infty$. $\qquad\square$

**Remark.** We note the mismatch in behavior between $\lim_{K\to\infty} M(K)$ and $\sum_{k=0}^{\infty}\|\nabla_\theta\, J_q(\pi_{\theta_k})\|_2^2$ also occurs outside of convergence regimes. For example, in the lower bound construction of Theorem 2 (see Section D), one readily deduces using (10) that $\|\nabla_\theta\, J_q(\pi_{\theta_k})\|_2$ vanishes exponentially fast so that $\sum_{k=0}^{\infty}\|\nabla_\theta\, J_q(\pi_{\theta_k})\|_2^2 < \infty$. At the same time, since there is a uniform gap and fixed step size, $M(K) \to \infty$ while the reward vanishes $J_q(K) \to 0$.

# G    FURTHER EXPERIMENTS AND DETAILS

## G.1    REINFORCE ON AN MAB PROBLEM

Here, we consider a simplified version of the experiment in Section 5.1, for a single context/prompt. We run REINFORCE (with an exact value baseline) on a tabular softmax policy class over a bandit problem with 100 arms, with a randomly chosen best arm, having reward 1 with all other arms 0. The logits $\theta_k$ are initialized as a constant and updated according to the exact gradient update rule:

$$\theta_{k+1} = \theta_k + \eta\cdot\mathbb{E}_{y\sim\pi_{\theta_k}}[(\nabla_\ell \log\pi_{\theta_k}(y))\cdot(r(y) - J(\pi_{\theta_k}))]\,,$$

with fixed stepsize $\eta > 0$, run over $K := 10,000$ steps.

We construct three plots in Figure 4, based on the value function $J(k)$, gradient gap $\Delta\mu(k)$, and the cumulative gradient gap $\sum_{k=0}^{K}[\Delta\mu(k)]_+ \cdot \eta_k$, colored by the step count $k$. Interestingly, the

curve $k \mapsto \Delta\mu_q(k)$ behaves as a concave quadratic. This is in fact explainable via calculating the alignment gap using our advantage expression (10):

$$\boldsymbol{w}_k = \nabla_\theta J(\pi_{\theta_k}) = (\pi_{\theta_k}(a^\star) \cdot (1 - \pi_{\theta_k}(a^\star)) \cdot \mathbb{1}\{a = a^\star\} - \pi_{\theta_k}(a) \cdot \pi_{\theta_k}(a^\star) \cdot \mathbb{1}\{a \neq a^\star\})_{a \in [100]} \, .$$

Thus,

$$\Delta\mu_q(k) = \frac{1}{J(\pi_{\theta_k}) \cdot (1 - J(\pi_{\theta_k}))} \langle \nabla_\theta J(\pi_{\theta_k}), \nabla_\theta J(\pi_{\theta_k}) \rangle$$

$$= \pi_{\theta_k}(a^\star) \cdot (1 - \pi_{\theta_k}(a^\star)) + \pi_{\theta_k}(a^\star) \sum_{a \neq a^\star} \frac{\pi_{\theta_k}(a)^2}{1 - \pi_{\theta_k}(a^\star)} \, .$$

Now, defining the vector $v := (\pi_{\theta_k}(a))_{a \neq a^\star}$, since we have $\frac{1}{99}\|v\|_1^2 \leq \|v\|_2^2 \leq \|v\|_1^2$ and since $\|v\|_1 = 1 - \pi_{\theta_k}(a^\star)$, we have the above RHS scales like $\approx \pi_{\theta_k}(a^\star) \cdot (1 - \pi_{\theta_k}(a^\star))$.

Altogether, our experiment here reinforces the message that increasing cumulative gradient gap corresponds to convergence as in Corollary 1.

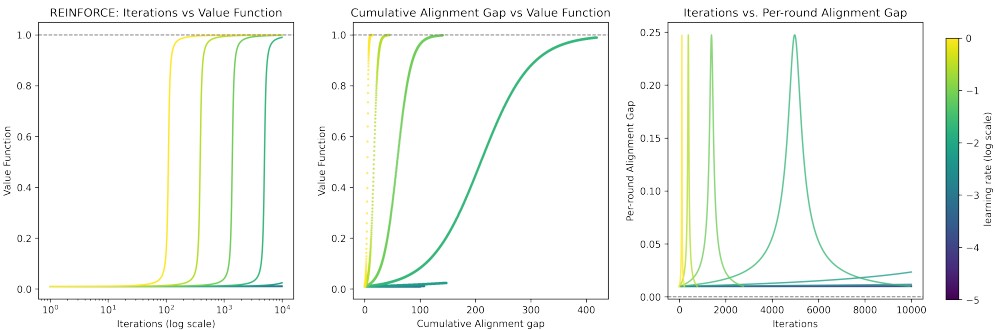

**Figure 4:** MAB Experiments.

## G.2 FURTHER ELABORATION ON CONTEXTUAL BANDIT EXPERIMENTS

Our theoretical analysis so far has focused on convergence for a single prompt $q$. A natural question is: how does the theory extend to the case of multiple prompts or questions? To illustrate this, we consider a contextual bandit simulation. In this setup, each iteration $k$ draws a random context $x_k$ (equivalent to a prompt $q_k$ in our framework), and the update gradient $\boldsymbol{w}_k$ is computed from that context $x_k$. Crucially, the same update direction is then applied globally to all prompts (via the shared parameter $\theta$).

Two scenarios can arise: **Case (a):** *Alignment.* For some prompts $q$ (or contexts $x$), the update direction $\boldsymbol{w}_k$ aligns closely with the Gradient Gap $\boldsymbol{g}_q^+ - \boldsymbol{g}_q^-$. In these cases, the Gap Alignment $\Delta\mu_q(k)$ is positive and large, leading to effective improvement. **Case (b):** *Misalignment.* For other prompts, the Gradient Gap is nearly orthogonal to $\boldsymbol{w}_k$. Here, $\Delta\mu_q(k)$ is small, and applying the same step size $\eta_k$ can cause overshooting, preventing performance gains.

Our simulation confirms this intuition: some contexts consistently overshoot and are difficult to improve, while those with stronger alignment improve steadily—the better the alignment, the greater the improvement. For overshooting contexts, there is no gradual accumulation of $\Delta\mu_q(k)$ toward improvement; instead, crossing into the overshooting region acts as a decisive threshold, after which performance collapses toward zero with no recovery.

## G.3 DETAILS ON LANGUAGE MODEL EXPERIMENTS

Each of our training routines were performed on a single NVIDIA H200 SXM GPU (with 141GB of VRAM), using the TRL framework (von Werra et al., 2020) for training GRPO without KL regularization.

For training on GSM8k, we performed 3000 training steps with each step involving 4 prompts per batch, 4 responses per prompt in the GRPO estimator, a learning rate of $5 \cdot 10^{-6}$, and maximum prompt and completion sequence lengths of 256.

For training on DAPO-17k, we performed $1051$ training steps with each step involving 4 prompts per batch, 4 responses per prompt, a learning rate of of $10^{-6}$, and max sequence lengths of $2048$. For DeepScaleR, we used the same configuration for 733 training steps.

To compute the validated accuracies, we held out $20\%$ of the questions in each dataset as a holdout test/validation set. Finally, we used a binary reward function on the exact match of the final answer, with validated accuracies in Figure 2 reported as an average over 5 tries for each step's model.

