_{\mathrm{old}} = \eta_k \cdot w_k$ with $\|w_k\|_2 \le 1$. We denote $\eta := \eta_k$ and $w := w_k$ for short.

We apply the bound from equation (46) in Lemma 4 to the random variable $(L_\theta - L_{\theta_{\mathrm{old}}})(q, \vec{o})$. By evaluating this for the distributions corresponding to the positive space, $\vec{o} \sim \pi_{\theta_{\mathrm{old}}}(\cdot \mid q, \mathcal{O}_q^+)$, and the negative space, $\vec{o} \sim \pi_{\theta_{\mathrm{old}}}(\cdot \mid q, \mathcal{O}_q^-)$, we find that

$$\left| \log\left(\frac{J_q}{1 - J_q}\right) - \log\left(\frac{J_q^{\mathrm{old}}}{1 - J_q^{\mathrm{old}}}\right) - T_1 \right| \le 4 \|L_\theta - L_{\theta_{\mathrm{old}}}\|_\infty^2, \tag{47}$$

where

$$T_1 := \left\{ \mathbb{E}_{\vec{o} \sim \pi_{\theta_{\mathrm{old}}}(\cdot \mid q, \mathcal{O}_q^+)} - \mathbb{E}_{\vec{o} \sim \pi_{\theta_{\mathrm{old}}}(\cdot \mid q, \mathcal{O}_q^-)} \right\} \left[ (L_\theta - L_{\theta_{\mathrm{old}}})(q, \vec{o}) \right].$$

To establish the final bound in Lemma 1, we need to prove two intermediate results: (i) $T_1 = \Delta\mu_q(\pi_{\theta_{\mathrm{old}}}) \cdot \eta + \mathcal{O}(\eta^2)$ and (ii) $\|L_\theta - L_{\theta_{\mathrm{old}}}\|_\infty = \mathcal{O}(\eta)$. Once we show these two relationships hold, the desired bound (28) follows directly by substituting them into inequality (47).

**Analyzing term $T_1$:** For term $T_1$, Assumption 1 on the smoothness of function $L_\theta$ implies that

$$\left| (L_\theta - L_{\theta_{\mathrm{old}}})(q, \vec{o}) - \langle \nabla_\theta L_{\theta_{\mathrm{old}}}(q, \vec{o}), \theta - \theta_{\mathrm{old}} \rangle \right| \le \frac{L_{\mathrm{o}}}{2} \|\theta - \theta_{\mathrm{old}}\|_2^2.$$

According to our optimization scheme $\theta = \theta_{\mathrm{old}} + \eta \cdot w$, it follows that

$$\left| (L_\theta - L_{\theta_{\mathrm{old}}})(q, \vec{o}) - \eta\{ w \cdot \nabla_\theta L_{\theta_{\mathrm{old}}}(q, \vec{o}) \} \right| \le \frac{L_{\mathrm{o}}}{2} \eta^2. \tag{48}$$

Recalling the definition (8) of vectors $g_q^+ = g_q^+(\pi_{\theta_{\mathrm{old}}})$ and $g_q^- = g_q^-(\pi_{\theta_{\mathrm{old}}})$, we find that

$$g_q^+ - g_q^- = \left\{ \mathbb{E}_{\pi_{\theta_{\mathrm{old}}}^+(\cdot \mid q)} - \mathbb{E}_{\pi_{\theta_{\mathrm{old}}}^-(\cdot \mid q)} \right\} \left[ \nabla_\theta L_{\theta_{\mathrm{old}}}(q, \vec{o}) \right].$$

Moreover, the definition (11) of gap $\Delta\mu_q$ leads to

$$\Delta\mu_q(\pi_{\theta_{\mathrm{old}}}) = \langle w, g_q^+ - g_q^- \rangle = \left\{ \mathbb{E}_{\pi_{\theta_{\mathrm{old}}}^+(\cdot \mid q)} - \mathbb{E}_{\pi_{\theta_{\mathrm{old}}}^-(\cdot \mid q)} \right\} \left[ w \cdot \nabla_\theta L_{\theta_{\mathrm{old}}}(q, \vec{o}) \right].$$

Therefore, taking expectations $\left\{ \mathbb{E}_{\pi_{\theta_{\mathrm{old}}}^+(\cdot \mid q)} - \mathbb{E}_{\vec{o} \sim \pi_{\theta_{\mathrm{old}}}^-(\cdot \mid q)} \right\}(\cdot)$ of the terms inside the absolute value of equation (48), we get

$$\left| T_1 - \Delta\mu_q(\pi_{\theta_{\mathrm{old}}}) \eta \right| \le \frac{L_{\mathrm{o}}}{2} \eta^2. \tag{49}$$

**Bounding norm $\|L_\theta - L_{\theta_{\mathrm{old}}}\|_\infty$:** From inequality (48), we also have

$$\|L_\theta - L_{\theta_{\mathrm{old}}}\|_\infty \le \eta \sup_{\vec{o} \in \mathcal{O}} \left\{ w \cdot \nabla_\theta L_{\theta_{\mathrm{old}}}(q, \vec{o}) \right\} + \frac{L_{\mathrm{o}}}{2} \eta^2$$

$$\le \eta \sup_{\vec{o} \in \mathcal{O}} \|\nabla_\theta L_{\theta_{\mathrm{old}}}(q, \vec{o})\|_2 + \frac{L_{\mathrm{o}}}{2} \eta^2 \le G_{\mathrm{o}} \eta + \frac{L_{\mathrm{o}}}{2} \eta^2. \tag{50}$$

**Finalizing the proof:** Combining our previous bounds (47), (49) and (50), we get

$$\left| \log\left(\frac{J_q}{1 - J_q}\right) - \log\left(\frac{J_q^{\mathrm{old}}}{1 - J_q^{\mathrm{old}}}\right) - \Delta\mu_q(\pi_{\theta_{\mathrm{old}}}) \eta \right|$$

$$\le 4 \|L_\theta - L_{\theta_{\mathrm{old}}}\|_\infty^2 + \left| T_1 - \Delta\mu_q(\pi_{\theta_{\mathrm{old}}}) \eta \right|$$

$$\le \left\{ (2 G_{\mathrm{o}} + L_{\mathrm{o}} \eta)^2 + \frac{L_{\mathrm{o}}}{2} \right\} \eta^2 \le (L_{\mathrm{o}} + 8 G_{\mathrm{

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

_{\mathrm{p}}^2 \eta^2 \cdot |\vec{\boldsymbol{o}}| \right\} \right] = \mathbb{E}\big[ \mathbf{M}_{|\vec{\boldsymbol{o}}|} \big] \ \leq \ 1 .$$

*Cauchy-Schwarz and $\psi_1$ Norm.* We now isolate $\mathbb{E}[e^X]$ using the Cauchy-Schwarz inequality:

$$\mathbb{E}[e^X] \leq \mathbb{E}\left[ \exp\left\{ 2\,X - 2\,G_{\mathrm{p}}^2 \eta^2 \cdot |\vec{\boldsymbol{o}}| \right\} \right]^{\frac{1}{2}} \mathbb{E}\left[ \exp\left\{ 2\,G_{\mathrm{p}}^2 \eta^2 \cdot |\vec{\boldsymbol{o}}| \right\} \right]^{\frac{1}{2}} \leq \mathbb{E}\left[ \exp\left\{ 2\,G_{\mathrm{p}}^2 \eta^2 \cdot |\vec{\boldsymbol{o}}| \right\} \right]^{\frac{1}{2}} ,$$

where the last step used our supermartingale bound. Taking the logarithm gives

$$\log \mathbb{E}[e^X] \ \leq \ \frac{1}{2} \log \mathbb{E}\left[ \exp\left\{ 2\,G_{\mathrm{p}}^2 \eta^2 \cdot |\vec{\boldsymbol{o}}| \right\} \right] .$$

Finally, we bound the remaining term using the sub-exponential ($\psi_1$-Orlicz) norm of the response length, $T_{\psi_1}$ from Assumption 3. By definition,

$$\mathbb{E}\big[ \exp\{|\vec{\boldsymbol{o}}|/T_{\psi_1}\} \big] \ \leq \ 2 .$$

Consider a function $\phi(\lambda) := \log \mathbb{E}[\exp(\lambda\,|\vec{\boldsymbol{o}}|)]$, which is convex with $\phi(0) = 0$. Under condition $0 \leq 2\,G_{\mathrm{p}}^2 \eta^2 \leq 1/T_{\psi_1}$, the convexity implies

$$\phi(2\,G_{\mathrm{p}}^2 \eta^2) \ \leq \ 2\,G_{\mathrm{p}}^2 \eta^2 T_{\psi_1} \cdot \phi(1/T_{\psi_1}) \ \leq \ 2 \log 2 \cdot G_{\mathrm{p}}^2 \eta^2 T_{\psi_1} .$$

It follows that

$$\log \mathbb{E}\left[ \exp\left\{ 2\,G_{\mathrm{p}}^2 \eta^2 \cdot |\vec{\boldsymbol{o}}| \right\} \right] \ = \ \phi(2\,G_{\mathrm{p}}^2 \eta^2) \ \leq \ 2\,G_{\mathrm{p}}^2 \eta^2 \cdot T_{\psi_1} .$$

Plugging this back into our bound for $\log \mathbb{E}[e^X]$ gives

$$\log \mathbb{E}[e^X] \ \leq \ G_{\mathrm{p}}^2 \eta^2 \cdot T_{\psi_1} .$$

This establishes inequality (57b) and completes the proof.

# F   FURTHER EXPERIMENTS AND DETAILS

## F.1   REINFORCE ON AN MAB PROBLEM