# OpenReview forum: "On the Optimization Dynamics of RLVR: Gradient Gap and Step Size Thresholds"
_ICLR.cc/2026/Conference — Submitted to ICLR 2026_

### Official Review · Reviewer_iz1a · 2025-10-28

**Soundness:** 2
**Presentation:** 3
**Contribution:** 3
**Rating:** 4
**Confidence:** 4

**Summary:**

The paper investigates the conditions for global convergence in reinforcement learning from verifiable reward for LLM use cases. With the specific reward structure in RLVR, i.e., a binary reward at the end of the trajectory, the paper introduces the concept of gradient gap, defined as the gradient of the value function divided by the variance of the reward. The paper uses an analysis similar to the classic optimization literature or the Lyapunov stability analysis to show that if the accumulative gradient alignment (analogous to the accumulative negative drift in Lyapunov analysis) blows up, the policy gradient would obtain global convergence. Experiments on contextual bandits and LLMs are provided to partially validate the theoretical claim.

**Strengths:**

1. The paper is quite well written and easy to follow for a reader with a minimum optimization background.

2. The analysis of the theorems is intriguing to me. The authors chose a potential function (a generalized Lyapunov function) in the form of $\log\frac{J(\pi_k)}{1-J(\pi_k)}$, related to the binary reward structure of the RLVR problem, which is quite rare but interesting in the literature of optimization analysis. This potential function leads to the exponential convergence in the paper.

3. The problem studied is of relevance, and the theoretical and empirical techniques, for the most part, are solid.

**Weaknesses:**

My main is in the regime where the theory is interesting. Both theorems are much more informative in the convergence case, which requires $M(K)$ to blow up to infinity. This is a very strong assumption to me, because in classic non-convex optimization for general functions $f$, we could also derive some relation similar to (16), and sum them up, we would obtain that the value improved compared to the initial point is upper bounded by the negative of cumulative gradient norm square, i.e., $f^* - f(x_0) \leq f(x_K) - f(x_0) \leq - \sum_{k=1}^K \|\nabla f(x_t)\|_2^2$. But the gradient norm cannot blow up because it has a natural fixed lower bound.

Similarly, in this paper, $J_q(K)$ is upper bounded by the optimal value $J(\pi*)$ of the best policy, which may not be $1$. If $J(\pi*) < 1$, then $M(K)$ cannot blow up, since if it blows up, you will have $J_q(K)$ converges to $1$, which is strictly larger than the optimal policy, and it leads to a contradiction. In this case, I believe you can only show $\pi_K$ converges to a stationary policy, but not $J_q(K)$ converges to $1$.

Still, studying the case with $J(\pi*) = 1$ is already interesting enough. I believe what the paper lacks is an in-depth study and discussion of this quantity $M(K)$ and on what circumstances it blows up. For example, are there cases with REINFORCE or GRPO where the cumulative inner product with the gradient converges, but $M(K)$ blows up? The best case is if we estimate $w_k$ to be exactly the gradient gap. Then, will the squared norm of the gradient gap diverge in almost all RLVR instances? This needs to be proved. In the presentation of the current version, I don't see clearly the difference between the role of the inner product with gradient gap in (11) and the role of the squared gradient norm, which is a pity.

I will raise my score if, and only if, I'm convinced by theoretical arguments that $M(K)$ blowing up could happen in most cases, or $M(K)$ blowing up is a strictly weaker notation than the squared gradient norm blowing up, i.e., there is a RLVR instance where $M(K)$ blows up, but the sum of squared gradient norm over time converges.

**Questions:**

See weaknesses

---

> ### Author Response · Authors · 2025-11-25
>
> Thank you for your careful reading and interesting questions!
>
> First, we believe the reviewer intended to write the well-known gradient ascent inequality for maximization of an $L$-smooth objective $f$ using stepsize $\eta < 1/L$:
>
> $$\frac{\eta}{2} \sum\_{k=0}^{K-1} \|\nabla f(x\_k) \|^2 \leq  f(x\_K) - f(x\_0) \leq f^\* - f(x\_0)$$
>
> where $L$ is the Lipschitz constant of the policy gradient. We agree such an inequality can be used to show the gradient norm goes to $0$ and to derive rates of improvement on $J\_q(K)$ using the inequality $J\_q(\pi\_K) - J\_q(0) \geq \frac{\eta}{2} \sum\_{k=0}^{K-1} \| \nabla\_{\theta} J\_q(\pi\_k)\|\_2^2$. However, without further assumptions on the $\| \nabla\_{\theta} J\_q(\pi\_k)\|\_2$, we would only be able to conclude using the relation (10) in our paper that:
>
> $$J\_q(\pi\_K) - J\_q(0) \geq \frac{\eta}{2} \sum\_{k=0}^{K-1} \| \nabla\_{\theta} J\_q(\pi\_k)\|\_2^2 \geq \frac{\eta}{2} \sum\_{k=0}^{K-1} \left( J\_q(\pi\_k) \cdot \{ 1 - J\_q(\pi\_k) \} \cdot [ \Delta \mu\_q(k) ]\_+ \right)^2$$
>
> The **difficulty** here is that this lower bound itself involves the changing rewards $J\_q(\pi\_k)$. To contrast, our analysis using the log-odds (see (16) in the paper) allows for a clean lower bound on $J\_q(K)$ free of the other reward values $J\_q(\pi\_k)$. In fact, our analysis reveals that the simpler sum $M(K) := \sum\_{k=0}^{K-1} \eta \cdot [\Delta\mu\_q(k)]\_+$ directly corresponds to improvement in the reward. We do not see how this fact is readily evident from the above inequality.
>
> > I will raise my score if, and only if, I'm convinced by theoretical arguments that $M(K)$ blowing up could happen in most cases, or $M(K)$ blowing up is a strictly weaker notation than the squared gradient norm blowing up, i.e., there is a RLVR instance where $M(K)$ blows up, but the sum of squared gradient norm over time converges.
>
> We find that, so long as the accuracy $J\_q(\pi\_K) \xrightarrow{K\to\infty} 1$, we will have $M(K) \to \infty$ meaning that **in all situations where there's convergence, $M(K)$ blows up**. Furthermore, for a uniform gradient gap (the setting of Corollary 1), $M(K) \to \infty$ while $\sum\_{k=1}^{\infty} \|\nabla J(\pi\_k) \|^2 < \infty$, showing divergence of cumulative alignment gap is a strictly weaker notion than the sum of gradient norms blowing up.
>
> Finally, we note that **there are even situations outside of convergence regimes where $M(K)\to\infty$ and $\sum\_{k=1}^{\infty} \| \nabla J\_q(k) \| \to \infty$ are not aligned**. For instance, within the lower bound construction of Theorem 2, one can show $\| \nabla J\_q(k) \| \to 0$ exponentially fast whereas $M(K) \to \infty$. Despite this, we have the catastrophic behavior $J\_q(K) \to 0$ because the stepsize is larger than the prescribed threshold.
>
> We've included these results with proofs in Appendix F (p. 26) of the current revised version (edits highlighted in blue). We hope this fully addresses your question.

---

> ### Comment · Reviewer_iz1a · 2025-11-27
>
> Thank you for your response. I've taken some time to think of an easy example where the assumptions in your Corollary 1 would be true. And it seems in a very simple example where you have only one question $q$, one good response and one bad response, with a softmax policy parameterization, we will have $g_q^+(\pi_{\theta}) - g_q^-(\pi_\theta) = 1$ independent of the parameter $\theta$. Then, to create a constant $\Delta \mu_q (k)$, you cannot naively take the $w_k$ to be the gradient $\nabla J_q(\pi_\theta)$, and you will need to take the normalized gradient so that the norm is $1$.
>
> The response from the authors and this example suffice to address my concerns. This indeed shows the special structure of the RLVR problem compared to classic smooth optimization and a deeper characterization. I have two suggestions for the authors to improve the paper:
>
> 1. Include a version of the stationary convergence in the paper's main text and some relevant discussion on how the message from Theorem. 1 differs from saying the following uninteresting sentence: "If my sum of gradient norm squares blows up, I will get global convergence." Explain that you characterized a finer relation.
>
> 2.  After presenting Corollary 1, use a toy example to explain that the uniform gradient gap is realistic. It could be a fact in, for example, a two-answer softmax parameterization. There are also interesting aspects you could discuss: for example, in this case, vanilla gradient descent is not enough to guarantee a uniform gradient gap.
>
> There are limitations to this paper. For example, the theory only characterizes clearly the case where the optimal policy could answer all questions correctly, i.e., $J(\pi^*)=1$. Overall, I feel there is more value for the paper to be accepted than rejected. I have adjusted my score.

---

### Official Review · Reviewer_ARfP · 2025-11-01

**Soundness:** 3
**Presentation:** 2
**Contribution:** 3
**Rating:** 6
**Confidence:** 3

**Summary:**

This paper establishes a theoretical foundation for RLVR by introducing the Gradient Gap concept, proving that convergence depends on aligning updates with this gap and deriving a critical step-size threshold that explains empirical heuristics like length normalization.

**Strengths:**

1. Current RLVR algorithms are mostly empirically based, and theoretical analysis is indeed a relatively underexplored area. How to train models stably is one of our key concerns.
2. The authors' theoretical analysis throughout the paper is quite rigorous, with step-by-step derivations that are convincing.
3. In Section 4, the authors analyze the strengths and weaknesses of state-of-the-art algorithms (such as GRPO and Dr. GRPO) based on the theory presented, demonstrating its practical relevance.

**Weaknesses:**

1. The manuscript requires further proofreading. For example, the section numbering of 1.1 Related Work is unusual, and in Section 5.2, the authors mention conducting two sets of GRPO experiments but claim to have conducted three. Such typos somewhat affect readability.
2. Some assumptions are overly idealized, revealing a gap between theory and practice.
3. This paper is almost purely theoretical, and the experimental validation is somewhat insufficient.

**Questions:**

1. In Equation 14, why is the $max(0, ·)$ operation applied to $\Delta\mu_q(k)$ ? If M(K) is intended to represent "useful progress," wouldn’t it be more reasonable for $\Delta\mu_q(k)$ to be negative when optimizing in the opposite direction? Additionally, could the authors briefly explain how M(K) was designed? Was it solely based on intuition?
2. In Section 4, the authors analyze the advantages and disadvantages of GRPO and Dr. GRPO. Is it possible for the authors to design a more effective algorithm based on the theory presented in this paper?

---

> ### Author Response · Authors · 2025-11-25
>
> Thank you for the careful review! We apologize for typos and have corrected them.
>
> **Weaknesses**
>
> > Some assumptions are overly idealized, revealing a gap between theory and practice.
>
> Our main assumptions (Assumptions 1 and 2) require boundedness and Lipschitz continuity of the policy score function, which are standard in theoretical analyses of policy gradient methods (e.g., "On the Theory of Policy Gradient Methods: Optimality, Approximation, and Distribution Shift", Agarwal et al., 2021). For LLM's with transformer architectures, these conditions hold on any compact region of parameter space. For LLM fine-tuning, weight decay keeps parameters bounded while gradient clipping directly enforces gradient norm constraints, suggesting training operates in such a region. Our experimental results on GRPO post-training of 7B models also align with the theoretical predictions, providing further evidence that the assumptions are reasonable in practice.
>
> > This paper is almost purely theoretical, and the experimental validation is somewhat insufficient.
>
> We've included an additional experiment post-training Qwen2.5-MATH-7B using GRPO on the DeepScaleR dataset. We're happy to provide further experiments if the reviewer would like to specify what further experiments would be more compelling.
>
> **Questions**
>
> > In Equation 14, why is the $\max(0,\cdot)$ operation applied to $\Delta\mu\_q(k)$? If $M(K)$ is intended to represent "useful progress", wouldn't it be more reasonable for $\Delta \mu\_q(k)$ to be negative when optimizing in the opposite direction? Additionally, could the authors briefly explain how $M(K)$ was designed? Was it solely based on intuition?
>
> In Theorem 1, part (b) on convergence, $\eta\_k$ is set to $0$ when $\Delta \mu\_q(k) \leq 0$ which ensures a "do no harm" stepsize preventing updates in the wrong direction. This allows us to capture benefits to the accuracy from nonnegative alignment gap. Nevertheless, a more general lower bound on the accuracy for arbitrary stepsizes holds involving possibly negative values of the alignment gap:
>
> $$J\_q(K) \geq \frac{J\_q(0)}{ J\_q(0) + (1-J\_q(0)) \cdot \exp \left( - \frac{1}{2} \sum\_{k=0}^{K-1} \Delta \mu\_q(k) \cdot \eta\_k - O(\eta\_k^2) \right) }$$
>
> > In Section 4, the authors analyze the advantages and disadvantages of GRPO and Dr. GRPO. Is it possible for the authors to design a more effective algorithm based on the theory presented in this paper?
>
> The implications of our stepsize choice are discussed on p. 7–8, Lines 370–399. From this, our theory recommends using a smaller stepsize choice within GRPO as $J\_q$ approaches $1$ in order to prevent stagnation in later stages of post-training.

---

> ### Comment · Reviewer_ARfP · 2025-11-28
>
> Thank you for the feedback! I will keep my initial score.

---

### Official Review · Reviewer_fwF7 · 2025-11-03

**Soundness:** 3
**Presentation:** 3
**Contribution:** 2
**Rating:** 4
**Confidence:** 2

**Summary:**

The paper provides a convergence analysis of RLVR with binary rewards and a single prompt.
It defines the notion of a "gradient gap", a rescaled version of the policy gradient, and derives convergence conditions based on it.
Finally experiments are conducted finetuning Qwen with GRPO.

**Strengths:**

* The convergence of RLVR methods is an interesting topic
 * I am not aware of a similar analyzes for RLVR
 * The presentation of the paper is overall clear

**Weaknesses:**

I will preface this by stating that **I do not work on theoretical analyses of policy gradient methods** and my comments are thus not very well informed.

* It seems the notion of "gradient gap" is simply a rescaled policy gradient, which the authors also state. It is introduced as the "improvement direction from low- to high-reward responses", which is exactly the policy gradient direction and well known. It is thus not clear to me why this is a useful new notion.
The paper also argues that "[gradient gap] is not scaled down by variability factor [...], making it a purer indicator of where to move [than the policy gradient]". However, it seems that for example gradient normalization would be a simpler way to achieve the same goal.
 * It is not clear from the draft which parts of the theoretical analysis are novel and which parts are based on prior work.
 * The theoretical analysis focuses on a single state/prompt per batch and thus neglects the group-wise advantage estimation of GRPO. However group-wise estimation is one main difference between GRPO and REINFORCE and would be useful to be discussed.

**Questions:**

* I would be grateful if the authors could explain further why the notion of gradient gap is necessary as opposed to using the normal policy gradient.

 * Most applications of RLVR use adaptive optimizers like Adam. It would be useful to discuss how this interacts with the step-size selection?

---

> ### Author Response · Authors · 2025-11-25
>
> Thank you for the careful review. We hope to clarify the difference between our work and policy gradient analysis here.
>
> **Weaknesses**
> > It seems the notion of "gradient gap" is simply a rescaled policy gradient... However, it seems that for example gradient normalization would be a simpler way to achieve the same goal.
>
> The reviewer is correct that the notion of gradient gap can be considered a rescaled policy gradient, as we say on p.4 using the identity (10). The main message of our paper is to validate algorithmic choices of popular approaches such as GRPO, Dr.GRPO, and REINFORCE, rather than to propose a new algorithm. Although "gradient normalization", as done by e.g. GRPO, is a widespread technique, there were previously **no known** rigorous policy gradient analyses of RLVR. Our main contribution is to present such a tight analysis (Theorems 3 and 4).
>
> Key in this analysis is our notion of gradient gap, which allows us to better highlight and justify the use of length-normalization in GRPO (the dependence on response length $T_{\infty}$ in Theorems 3 and 4) and also unify and contrast different normalization schemes (involving e.g. the variance of estimated gradients and response length) between GRPO and Dr.GRPO (see discussion at top of p. 8 which includes implications for empirically observed phenomena). Indeed, our worst-case construction in Theorem 3 shows that the stepsize must scale with the alignment gap to ensure convergence.
>
> > It is not clear from the draft which parts of the theoretical analysis are novel and which parts are based on prior work.
>
> Again, our analysis is completely novel and we are not aware of any other works, including theoretical works on policy gradient methods, which rigorously analyze stepsize choice and optimization dynamics for RLVR.
>
> > The theoretical analysis focuses on a single state/prompt per batch and thus neglects the group-wise advantage estimation of GRPO. However group-wise estimation is one main difference between GRPO and REINFORCE and would be useful to be discussed.
>
> The reviewer is correct that our theoretical results in Sections 3 and 4 focus on optimizing the accuracy of a single prompt. We note, however, that **our theory is independent of the choice of algorithm** and applies flexibly to any policy gradient approach, including those using group-wise estimation. This allows us to contrast the different gradient estimation choices made by GRPO vs. Dr.GRPO (see discussion on Lines 370–399).
>
> We also note that our experiments (Section 5) apply GRPO to the multi-prompt setting using group-wise estimation. For this broader setting, we see that the cumulative gap alignment $M(K)$, defined on a group level using batch-average quantities $\mathbb{E}_q[ \Delta \mu_q]$, captures the improvement or stagnation in validated accuracy, suggesting the theory holds at a group-level as well.
>
> **Questions**
>
> > I would be grateful if the authors could explain further why the notion of gradient gap is necessary as opposed to using the normal policy gradient.
>
> As stated above, there is no known policy gradient analysis of RLVR to our knowledge. Our theory and notion of gradient gap captures more refined dependence on response length and variance normalization, and tightly controls the step size threshold as shown by our upper and lower bounds (Theorems 3 and 4). Please also see our response to Reviewer iz1a for more discussion on why a classical gradient ascent analysis involving the usual policy gradient quantity $\| \nabla_{\theta} J_q(\pi_{\theta})\|$ poses challenges.
>
> > Most applications of RLVR use adaptive optimizers like Adam. It would be useful to discuss how this interacts with the step-size selection?
>
> Our notion of the gradient gap, and the stability conditions derived from it, apply to any policy gradient method, including Adam-style updates. In particular, we **do not assume** the updates to be exactly directionally aligned with the policy gradient.
>
> In fact, this generality is reflected in our language-model experiments (i.e., GRPO post-training of 7B models) which use the default AdamW optimizer of the TRL library and validate the theoretically proposed relationship between the cumulative gap alignment $M(K)$ and improvements in accuracy.

---

### Meta-Review · Area_Chair_m1s2 · 2026-01-08

**Summary:**

While the paper addresses an important topic——the convergence of RLVR methods. However, the novelty is limited, and the paper fails to adequately demonstrate its unique value compared to existing methods (Reviewer fwF7). The theoretical analysis rests on strong assumptions (Reviewers ARfP and iz1a), is detached from the experimental setup, and does not effectively guide practical algorithm design (e.g., the selection of learning rates). Multiple reviewers (Reviewer fwF7, Reviewer iz1a, Reviewer ARfP) have pointed out a significant gap between theory and experiment, and these key theoretical limitations remain unresolved in the authors' response. Therefore, the paper in its current form does not meet the acceptance criteria.

**Reviewer Concerns:**

The issues regarding the presentations have been clarified; however, the reviewers' primary concern—a gap between theory and practice—remains unaddressed.

**Reviewer Scores:**

Reviewer fwF7: The limitations of the theoretical analysis and the discussion regarding adaptive optimizers such as Adam have not been fully addressed. It is not expected that their score would improve.

Reviewer ARfP: The reviewer explicitly stated they would maintain their initial score.Reviewer iz1a: Although the points raised by this reviewer were not fully addressed, they are willing to revise their score.

---

### Decision · Program_Chairs · 2026-01-26

Reject